# Modelling proteins' hidden conformations to predict antibiotic resistance

Kathryn M. Hart[1], Chris M.W. Ho[1], Supratik Dutta[2], Michael L. Gross[2] & Gregory R. Bowman[1,3]

TEM β-lactamase confers bacteria with resistance to many antibiotics and rapidly evolves activity against new drugs. However, functional changes are not easily explained by differences in crystal structures. We employ Markov state models to identify hidden conformations and explore their role in determining TEM's specificity. We integrate these models with existing drug-design tools to create a new technique, called Boltzmann docking, which better predicts TEM specificity by accounting for conformational heterogeneity. Using our MSMs, we identify hidden states whose populations correlate with activity against cefotaxime. To experimentally detect our predicted hidden states, we use rapid mass spectrometric footprinting and confirm our models' prediction that increased cefotaxime activity correlates with reduced $\Omega$-loop flexibility. Finally, we design novel variants to stabilize the hidden cefotaximase states, and find their populations predict activity against cefotaxime *in vitro* and *in vivo*. Therefore, we expect this framework to have numerous applications in drug and protein design.

[1] Department of Biochemistry & Molecular Biophysics, Washington University School of Medicine, 660 South Euclid Avenue, St Louis, Missouri 63110, USA. [2] Department of Chemistry, Washington University in St Louis, One Brookings Drive, St Louis, Missouri 63130, USA. [3] Department of Biomedical Engineering, and Center for Biological Systems Engineering, Washington University in St Louis, One Brookings Drive, St Louis, Missouri 63130, USA. Correspondence and requests for materials should be addressed to G.R.B. (email: bowman@biochem.wustl.edu).

Antibiotic resistance is a global health threat that results in millions of deaths and billions of dollars in health-care costs every year[1]. Expression of the enzyme TEM β-lactamase (TEM) is the predominant mechanism underlying antibiotic resistance in pathogenic Gram-negative bacteria[2]. TEM quickly evolves the ability to degrade new drugs as they are introduced in the clinic, but how changes in sequence alter this protein's specificity remains a mystery despite decades of structural and biochemical research[3].

Unlike enzymes where conformational changes are known to be important for function[4], TEM is thought to be quite rigid[5]. Therefore, many of the models proposed to explain how mutations alter specificity focus on the possibility that the substituted residues interact with the substrate[6–8]. However, many of these substitutions are too far from the active site to be involved in direct interactions[3]. Moreover, it is challenging to explain the effects of many mutations in terms of structural changes, as the differences between crystal structures of TEM variants with dramatically different specificities are extremely subtle (Fig. 1). For example, TEM-52 (E104K/G238S/M182T) hydrolyzes a third-generation cephalosporin, cefotaxime, 2,300-fold more efficiently than TEM-1 (ref. 9). Although there are some notable conformational changes in loops flanking the active site (Supplementary Fig. 1)[7,10], the active-site residues themselves are essentially identical (Fig. 1 inset, r.m.s.d. = 0.33 Å).

We hypothesize that hidden conformations not yet captured by traditional structural techniques are the missing ingredients required to connect TEM's structure with function and to predict the effects of mutations. This hypothesis is supported by computational models and room temperature crystals that have revealed TEM adopts diverse structures[11–14]. A growing body of work argues for the importance of conformational heterogeneity in processes like allostery[15–17], ligand binding[18–22] and catalysis[4,23,24]. Unfortunately, it remains difficult to make a direct link between conformational heterogeneity and function.

Here, we employ Markov state models (MSMs)[25] to explore the role of conformational heterogeneity in TEM β-lactamase activity. An MSM is essentially a map of the ensemble of structures that a protein adopts. These models are constructed using atomically detailed molecular dynamics simulations to identify the structural states a protein populates, their equilibrium probabilities and the rates of transitioning between them. MSMs have proven a powerful means to understand many biomolecular processes[26,27], and there are now powerful methods for constructing these models[28–30]. The combination of thermodynamic, kinetic and structural information from MSMs can provide mechanistic insight and guide the application of experimental techniques[23,24,31] for identifying hidden conformations. We reasoned MSMs could reveal hidden conformations that determine TEM specificity but that are not apparent from single 'snap-shot' structures. First, we integrate our MSM with computational docking to develop a new approach called Boltzmann docking. Then, we test the MSM's predictions of conformational heterogeneity using fast photochemical oxidation of proteins (FPOP), a chemical-footprinting technique. Finally, to determine the importance of hidden states, we design new variants to control their populations and then measure their activities *in vitro* and *in vivo*.

## Results

**Docking against one structure fails to predict activity.** If the ability of ground-state structures of TEM variants to bind different substrates specifies their activities, then it should be possible to make predictions by docking substrates against crystal structures, when available, or homology models when structures have not yet been solved. To test this hypothesis, we chose to study a series of variants with differing activities against cefotaxime, a substrate for which new activity evolved in the clinic. First, we built homology models for the variants. Although there are crystal structures for many different variants of TEM, this is not true for other enzymes we might wish to study. To preserve the generalizability of this method, we chose to build all the homology models using TEM-1 (ref. 10) to mimic situations where only one structure might be available. Then, we docked cefotaxime against the active sites of each of these variants. Because we use substrate docking as a proxy for activity, we expect higher scores to predict greater activity against the docked substrate.

Docking against single structures of each variant fails to predict their abilities to degrade cefotaxime, as measured by their $k_{cat}/K_m$ values. In fact, we observe that the docking scores and activities are anticorrelated ($R = -0.37 \pm 0.07$, Fig. 2a). While it is possible this anticorrelation suggests an alternative model for TEM catalysis, it is more likely that it simply highlights the limitations of docking against single structures. For example, this model would incorrectly predict that compounds with no binding affinity would be excellent substrates. Examining our homology models of variants containing the substitution G238S revealed that they do not capture a subtle shift in the 238-loop (residues 238–242) that has been observed in a number of crystal structures of variants with this substitution. To check that failure to capture this subtle shift is not responsible for the poor correlation in Fig. 2a, we repeated the experiment using a TEM-52 crystal structure[7] (E104K/G238S/M182T) as the template for all variants containing the G238S substitution. However, this alternative protocol did little to improve the prediction, again showing an anticorrelation between docking and activity ($R = -0.30 \pm 0.05$). These failed predictions could be due to shortcomings in the force field, which describes the atomic interactions used to produce a docking score. However, they could also be

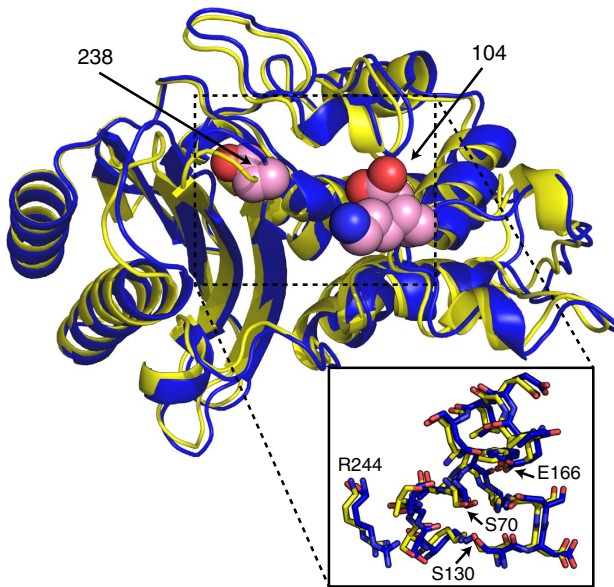

**Figure 1 | Structural comparison of TEM β-lactamases with differing substrate specificities.** Overlay of TEM-1 (blue, PDB 1BTL) and TEM-52 (yellow, PDB 1HTZ) reveals subtle conformational differences (heavy atom r.m.s.d. = 0.60 Å), particularly in the loops containing the mutations, but very similar active-site architectures (inset, heavy atom r.m.s.d. = 0.33 Å). Residues at positions 104 and 238 flank the active site and are shown in pink spheres. See also Supplementary Fig. 1.

interpreted as evidence for the inadequacy of focusing on single, rigid structures when we know that proteins actually adopt a distribution of different structures at thermal equilibrium. Indeed, we have previously demonstrated that TEM β-lactamase adopts a range of different conformations[12,32]. Therefore, we wanted to explore whether inadequacies in the force field are really to blame for the failures of docking, or if inadequate accounting for proteins' hidden states is responsible.

**Boltzmann docking improves activity predictions.** To test whether considering proteins' hidden states is crucial for predicting their functions, we developed a technique called Boltzmann docking that approximates compounds' relative binding affinities by calculating the ensemble-average score across a set of structural states, weighting each state by its equilibrium probability. Our approach differs from existing ensemble docking methods, which dock compounds against a set of structures sampled via molecular dynamics simulations and then rank the compounds based on the highest score against any of the target structures[33–35]. Boltzmann docking is more similar to methods that dock compounds against multiple conformations from NMR or crystal structures and weight the score against each structure by their relative contribution to the experimental signal[36].

To perform Boltzmann docking, we first built an MSM for each variant based on 2.5 μs of atomically detailed molecular dynamics simulations. Then we docked cefotaxime against a representative structure from each state and calculated an average score, weighting the contribution of each state by its equilibrium probability according to the MSM. More details are given in Methods section.

Boltzmann docking of cefotaxime against each of our variants represents a vast improvement over docking against single structures (Fig. 2b, $R = 0.30 \pm 0.10$ versus $-0.37 \pm 0.07$). For comparison, we also performed classical ensemble docking, which ranks compounds based on their highest score against any single conformation from a set of structures. We find a correlation coefficient between ensemble docking and experiments of $R = -0.02 \pm 0.07$, indicating no correlation. Thus, while ensemble docking is superior to docking against single structures, which shows an anticorrelation, population-based averaging with Boltzmann docking is the only method that results in a physically meaningful correlation with experiments. Interestingly, the highest scoring docking poses for both wild type and E104K/G238S bind to structures that closely resemble the ligand-free crystal structures (Fig. 3). This observation stands in contrast to arguments that the active site must open up to accommodate larger substrates, such as cefotaxime[7,11,37], as discussed in more detail below.

These results suggest Boltzmann docking has the potential to predict activity against substrates based only on their chemical structure, and thus anticipate resistance to new antibiotics. Our results also suggest current force fields are better than many have inferred from docking against single structures. However, realizing the potential of current force fields requires a proper

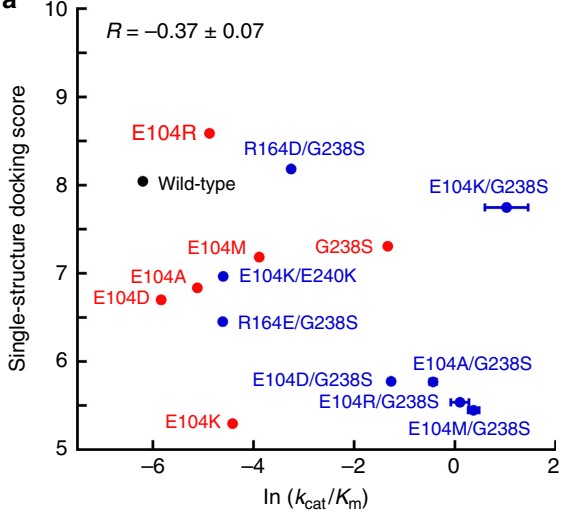

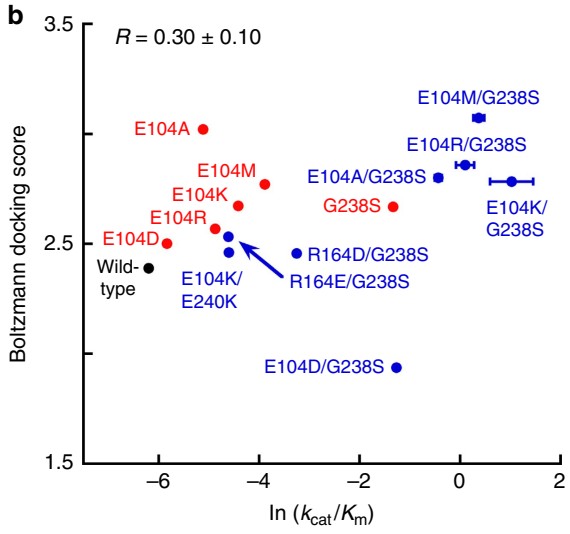

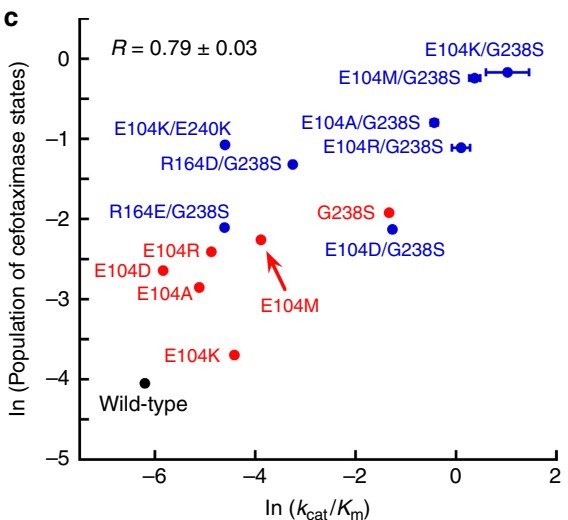

**Figure 2 | An ensemble perspective predicts the effects of mutations on TEM β-lactamase's specificity better than single structures.** (**a**) Docking scores for cefotaxime against single-structure homology models of each variant are anticorrelated ($R = -0.37 \pm 0.07$) with the measured catalytic efficiencies ($\ln(k_{cat}/K_m)$). (**b**) There is a correlation ($R = 0.30 \pm 0.10$) between the Boltzmann docking scores for cefotaxime and the measured catalytic efficiencies ($\ln(k_{cat}/K_m)$). (**c**) There is a stronger correlation ($R = 0.79 \pm 0.03$) between the natural logarithm of the populations of cefotaximase states and the measured catalytic efficiencies ($\ln(k_{cat}/K_m)$). The correlation is robust to exclusion of E104K/G238S ($R = 0.74 \pm 0.04$). Double mutants are shown in blue, and single mutants are shown in red. Error bars are standard errors from the fit. Computational errors are less than $3 \times 10^{-3}$. The natural logarithms of MSM populations and experimentally determined catalytic efficiencies are reported to put these quantities on an energy scale, while docking scores are naturally on an energy scale.

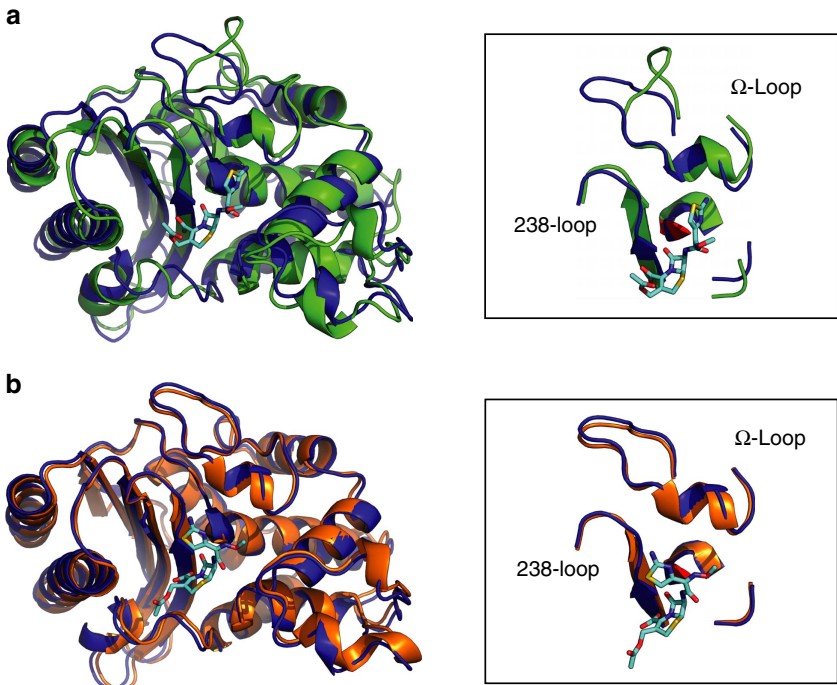

**Figure 3 | Bound states from Boltzmann docking.** Highest ranking poses from Boltzmann docking of cefotaxime against (**a**) wild type (green) and (**b**) E104K/G238S (orange). Both structures are shown overlaid with the crystal structure of TEM-1 (blue, PDB 1BTL) for reference. Cefotaxime is in cyan. Insets depict key loops flanking the active site with the catalytic Ser70 shown in red.

accounting for proteins' conformational heterogeneity. Future efforts to account for how protein–ligand interactions redistribute the equilibrium probabilities of different structures could lead to even further improvements without requiring any alteration of the underlying force field. Despite the apparent trend in correlation coefficients as one moves from docking against single structures, to ensemble docking, and then to Boltzmann docking, we acknowledge that we cannot discount the possibility that this trend is due to chance, given the number of data points we have[38]. Furthermore, while Boltzmann docking appears to outperform alternative methods, the absolute correlation between Boltzmann docking and experiments is only moderate. It does suggest, however, that our MSMs contain information that is not encoded in single structures. We reasoned that our MSMs could contain information about the catalytic cycle beyond just the substrate-binding affinity, and next sought to learn what insights further analysis of these models might provide.

**MSMs identify hidden cefotaximase states.** To determine which conformational states are responsible for changes in substrate specificity, we compared a cefotaxime-degrading variant, E104K/G238S, with a wild-type reference, TEM-1. E104K/G238S hydrolyzes cefotaxime 1,400-fold more efficiently than wild type, and *Escherichia coli* expressing this variant have a >500-fold increase in their minimum inhibitory concentration (MIC) (Table 1). It was suggested that the G238S substitution was acquired first during evolution, because this single variant has an MIC of 1.13 μM, whereas E104K alone has little effect[39,40]. Numerous models have been proposed for how these substitutions alter TEM's specificity. Two proposals are that they form direct interactions with the oxyimino group of third-generation cephalosporins, and that G238S opens up the active site to better accommodate the larger substrates[3,6–8,37]. However, none of these models provides a quantitative means to predict the activities of new variants, nor do they account for the

hidden states we have shown are important determinants of TEM's specificity.

To determine how the E104K/G238S substitutions alter the specificity of TEM, we constructed MSMs for both wild type and E104K/G238S. We used the crystal structure of wild type[10] and a homology model of E104K/G238S as starting points for 2.5 μs of explicit solvent molecular dynamics simulations per sequence. We pooled the two datasets together and used MSMBuilder[28] to cluster them based on the r.m.s.d. of shared residues in the active site and then determined the equilibrium thermodynamics and kinetics of each sequence independently in this shared state space.

To identify states that may be responsible for degrading cefotaxime, which we call cefotaximase states, we queried the MSMs for all states that are significantly more populated by the E104K/G238S variant, which can degrade both cefotaxime and benzylpenicillin, over wild type, which can only degrade benzylpenicillin effectively. We also identified structures that are more populated by wild type than E104K/G238S, which we refer to as non-cefotaximase states. This analysis reveals that the Ω-loop of wild type undergoes substantial rearrangements that are absent in E104K/G238S (Fig. 4). The Ω-loop, comprising residues 164–179 (ref. 41), is of known importance. It interacts directly with the substrate[3], helps to coordinate a water required for catalysis[42], and is extremely sensitive to mutation[41]. In cefotaximase states, the Ω-loop conformation closely resembles the crystal structures of TEM-1 and TEM-52 (Fig. 1), whereas in non-cefotaximase states, the Ω-loop extends away from the active site (Fig. 4). This result contrasts with crystallographic studies that claim widening the active site, via motion in a different loop, is important for binding bulky substrates like cefotaxime[7,11]. They observe movement of the 238-loop (residues 238–242) away from the Ω-loop and attribute it to loss of a key hydrogen bond between the backbone amides of Glu240 and Asn170 in cefotaximase variants. We do not observe this motion in our most populated cefotaximase states (Fig. 4c), as discussed in more detail below.

**Table 1 | *In vitro* and *in vivo* activities of TEM β-lactamase variants\*.**

| | Benzylpenicillin | | | | Cefotaxime | | | |
|---|---|---|---|---|---|---|---|---|
| | $k_{cat}$ (s$^{-1}$) | $K_m$ (µM) | $k_{cat}/K_m$ (µM$^{-1}$ s$^{-1}$) | MIC (mM)$^†$ | $k_{cat}$ (s$^{-1}$) | $K_m$ (µM) | $k_{cat}/K_m$ (µM$^{-1}$ s$^{-1}$) | MIC (µM)$^†$ |
| TEM-1 | 1,300 ± 50 | 35 ± 4 | 37 ± 5 | 24 | ND$^‡$ | ND$^‡$ | $2.0 \times 10^{-3} \pm 0.5 \times 10^{-4}$ | <0.035 |
| M182T | 780 ± 40 | 21 ± 4 | 38 ± 7 | ND | ND$^‡$ | ND$^‡$ | $1.8 \times 10^{-3} \pm 0.2 \times 10^{-4}$ | 0.07 |
| G238S | 66 ± 1 | 4.3 ± 0.4 | 16 ± 2 | 12 | 50 ± 3 | 190 ± 20 | 0.26 ± 0.03 | 1.13 |
| E104K | 1,200 ± 60 | 39 ± 6 | 30 ± 5 | 24 | ND$^‡$ | ND$^‡$ | $1.2 \times 10^{-2} \pm 0.4 \times 10^{-3}$ | 0.07 |
| E104K/G238S | 38 ± 2 | 2.3 ± 0.5 | 17 ± 4 | 12 | 87 ± 4 | 31 ± 5 | 2.8 ± 0.4 | 18 |
| E104R | 970 ± 40 | 60 ± 6 | 16 ± 2 | 24 | ND$^‡$ | ND$^‡$ | $7.6 \times 10^{-3} \pm 0.1 \times 10^{-4}$ | 0.07 |
| E104R/G238S | 25 ± 2 | 3.6 ± 1.4 | 7.1 ± 2.9 | 12 | 38 ± 2 | 34 ± 5 | 1.1 ± 0.2 | 4.5 |
| E014A | 1,200 ± 70 | 30 ± 5 | 41 ± 7 | 24 | ND$^‡$ | ND$^‡$ | $6.0 \times 10^{-3} \pm 0.4 \times 10^{-3}$ | <0.035 |
| E104A/G238S | 52 ± 4 | 3.5 ± 1.1 | 15 ± 5 | 6 | 47 ± 2 | 72 ± 7 | 0.65 ± 0.07 | 2.25 |
| E104D | 1,700 ± 200 | 190 ± 40 | 8.8 ± 2.4 | 24 | ND$^‡$ | ND$^‡$ | $2.9 \times 10^{-3} \pm 0.5 \times 10^{-4}$ | <0.035 |
| E104D/G238S | 45 ± 1 | 2.9 ± 0.6 | 15 ± 3 | 6 | 57 ± 5 | 200 ± 30 | 0.28 ± 0.04 | 0.56 |
| E104M | 1,100 ± 40 | 14 ± 2 | 78 ± 11 | 24 | 4.8 ± 0.5 | 230 ± 30 | $2.1 \times 10^{-2} \pm 0.4 \times 10^{-2}$ | 0.07 |
| E104M/G238S | 110 ± 8 | 15 ± 4 | 7.1 ± 1.9 | 12 | 78 ± 2 | 53 ± 4 | 1.5 ± 0.1 | 9 |
| E104I | 1,200 ± 70 | 28 ± 4 | 45 ± 8 | 24 | ND$^‡$ | ND$^‡$ | $8.7 \times 10^{-3} \pm 0.1 \times 10^{-3}$ | <0.035 |
| E104I/G238S | 66 ± 3 | 5.9 ± 1.5 | 11 ± 3 | 12 | 89 ± 4 | 56 ± 7 | 1.6 ± 0.2 | 18 |
| E240K/E104K | 1,300 ± 50 | 57 ± 6 | 24 ± 3 | 24 | ND$^‡$ | ND$^‡$ | $1.0 \times 10^{-2} \pm 0.3 \times 10^{-3}$ | <0.035 |
| R164E/G238S | 3.5 ± 0.1 | 36 ± 4 | 0.10 ± 0.01 | 0.19 | 0.7 ± 0.1 | 66 ± 13 | $9.9 \times 10^{-3} \pm 2.1 \times 10^{-3}$ | <0.035 |
| R164D/G238S | 9.4 ± 0.2 | 9.5 ± 1.1 | 1.0 ± 0.1 | 1.5 | 4.8 ± 0.4 | 123 ± 19 | $3.9 \times 10^{-2} \pm 0.7 \times 10^{-3}$ | <0.035 |

MIC, minimum inhibitory concentration; ND, not determined.
\*Standard error values from the fits are reported for $k_{cat}$ and $K_m$. MIC determination was repeated at least three times. Values are most commonly observed concentration with an error of $+/-$ one well.
†The *E. coli* strain used here (DH5α) has an intrinsic resistance of 0.05 mM for benzypenicillin and <0.035 µM for cefotaxime. MICs were also measured in BL21(DE3) cells, and similar trends were observed.
‡Not determined. Michaelis–Menten curve did not saturate. $k_{cat}/K_m$ was determined by a linear fit.

If the conformational preferences and amplitudes of the Ω-loop's fluctuations are key determinants of the different specificities, then we should be able to predict the activities of other variants based on the properties of their Ω-loops. Variants that populate the cefotaximase states should have higher activities against this substrate than variants that preferentially populate the non-cefotaximase states. Because the single G238S substitution confers substantial resistance to cefotaxime but the E104K substitution does not, we expect G238S to resemble E104K/G238S, whereas E104K should more closely resemble wild type. Consistent with our hypothesis, the G238S substitution populates the cefotaximase states more than wild type and E104K but less than E104K/G238S (Fig. 2c). E104K populates the cefotaximase states to about the same extent as wild type, consistent with the minimal impact this single substitution has on cefotaxime activity.

Examining the distribution of structures reveals that each substitution pins down the side of the Ω-loop to which it is adjacent. G238S appears to do this by hydrogen bonding with residues in the Ω-loop. For example, it forms a hydrogen bond with the carbonyl of Asn170 in 70% of the population (Supplementary Fig. 2b). E104K also increases interactions with the Ω-loop, as shown by a decreased distance between position 104 and Pro167 relative to wild type (Supplementary Fig. 2a). Interestingly, although the charge change has previously been cited as the basis for rate enhancement[8], our observation that E104K packs against Pro167 suggests van der Waals contacts with the Ω-loop also play a role. Pinning down both sides of the Ω-loop leads to the large reduction in Ω-loop heterogeneity in the double mutant and correlates with increased rates of cefotaxime hydrolysis. This observation runs counter to a common assumption that more promiscuous enzymes have greater heterogeneity in their active sites[43].

Both of these substitutions have been studied extensively, but to our knowledge we are the first to present a model linking either mutation to restricted motion in the Ω-loop. Crystal structures of G238S-containing variants present conflicting models for how the 238-loop and Ω-loop interact. The TEM-52 structure lacks

hydrogen bonds between the two loops[7], but other structures capturing multiple conformations of the Ser238 side chain show it can hydrogen bond to the backbone carbonyls of Asn170 or Glu171 (ref. 11). We also observe formation of a hydrogen bond between Ser238 side chain and Asn170 backbone, but the precise conformation of the 238-loop differs. The crystal structures show that the 238-loop moves 2–4 Å away from the Ω-loop (measured by the $C_α$ positions of Glu240 in G238S-containing structures versus TEM-1), which results in a small widening of the active site. In contrast, we observe that both loops assume conformations more similar to the wild-type crystal structure in our most populated cefotaximase states (Fig. 4). Our model suggests that rather than opening up the active site, G238S and E104K maintain a closed active-site architecture by pinning down the Ω-loop. In the absence of these substitutions, the Ω-loop exhibits greater flexibility and accesses conformations that increase its solvent exposure. This opening event in wild type is a more dramatic change than the small widening caused by movement of the 238-loop in G238S-containing crystal structures. The models are not necessarily mutually exclusive, but ours challenges the proposal that wild type cannot degrade cefotaxime because it is simply too big to fit in the active site. Our Boltzmann docking analysis supports our model because the structures with the highest docking score against cefotaxime are similar between wild type and E104K/G238S (Fig. 3). Both docked structures have loop conformations resembling the wild-type crystal structure (Fig. 3, insets). We do not observe active-site opening when cefotaxime is bound, and the larger substrate can, indeed, fit in the wild-type active site without steric conflict. It is the fact that this binding-competent state is not highly populated in wild type that underlies its low activity against cefotaxime. Finally, kinetic analysis of cefotaximase variants shows that higher catalytic efficiencies likely result from lower $K_m$ values. Because acylation is the rate-limiting step, $K_m$ approximates the substrate-binding affinity[44]. Cefotaximase variants also have lower $K_m$s for benzylpenicillin (Table 1), which suggests restricting motion in the Ω-loop helps the enzyme bind both substrates. The low $k_{cat}$ values observed for these variants

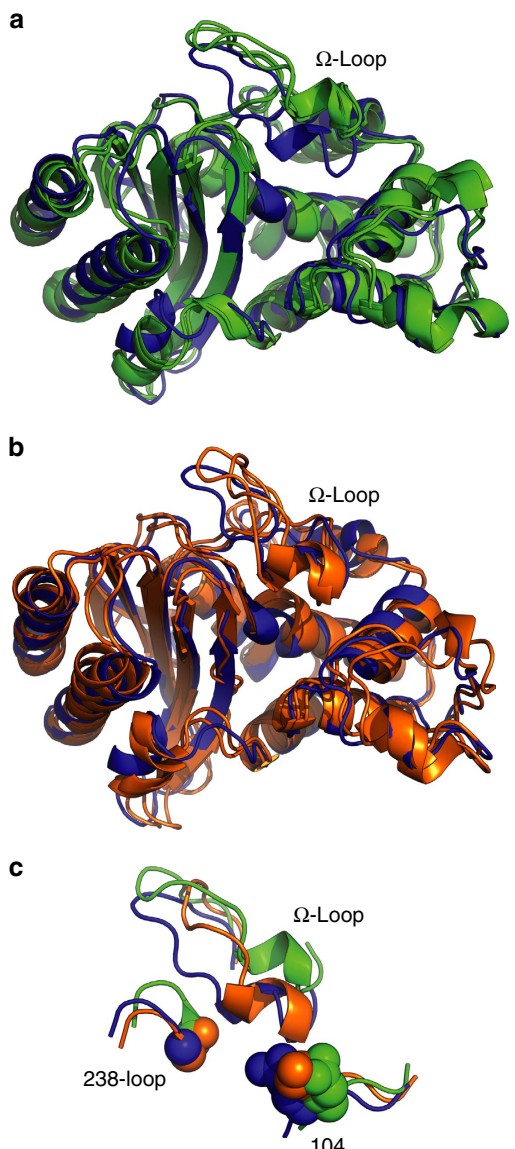

**Figure 4 | Functionally relevant states from the MSMs.** A crystal structure of TEM-1 (blue, PDB 1BTL) is overlaid with the two most populated structures taken from MSMs of the (**a**) non-cefotaximase states (green), which are favoured by wild type and the (**b**) cefotaximase states (orange), which are favoured by the E104K/G238S. (**c**) Large structural rearrangements in the Ω-loop distinguish low-energy non-cefotaximase states from cefotaximase states, which more closely resemble the conformation captured by the crystallographic structure. Residues 104 and 238 are shown in spheres.

might indicate that Ω-loop flexibility is important for another step in the catalytic cycle, such as product release. We propose that the conformation captured by the wild-type crystal structure, particularly of the Ω-loop, is the binding-competent state for a diverse set of substrates, and that E104K/G238S stabilizes this state to allow for more promiscuous binding.

**Footprinting confirms Ω-loop rigidity in cefotaximases.** Because the Ω-loop mobility we observe in MSMs results in significant changes in solvent accessibility (Fig. 4), we can experimentally test our insights using the FPOP approach. This technique is a chemical-footprinting method that reports on

structural fluctuations by labelling solvent-exposed side chains with hydroxyl radicals and detecting the oxidized peptides with mass spectrometry[45]. We previously found that FPOP labelling distinguishes a flexible loop in ApoE, whose motions are invisible to slower hydrogen deuterium exchange and GEE labelling[46], and have used it to follow fast folding of barstar protein on the millisecond timescale[47]. Labelling with primary radicals occurs on the microsecond timescale, which is much faster than TEM unfolding[48]. This labelling rate is also similar to the timescale of Ω-loop motions in wild type and faster than Ω-loop motions in E104K/G238S. Therefore, labelling reports on the solvent accessibility of conformations in the folded state, making FPOP a powerful way to assess the MSM prediction that reduced Ω-loop heterogeneity correlates with increased cefotaxime hydrolysis.

The effects of the E104K and G238S substitutions were evaluated in the background of the well-characterized stabilizing mutation M182T (ref. 49) to aid in data collection. The M182T substitution alone stabilizes TEM but has little effect on its activity (Table 1). The triple mutant E104K/G238S/M182T is found clinically and, like the double mutant, exhibits cefotaximase activity[50]. We compared the difference in labelling between E104K/G238S/M182T and M182T and observe reduced labelling at a number of sites in the triple mutant, particularly the region preceding the Ω-loop and the Ω-loop itself (Fig. 5). Interestingly, loss of mobility due to the E104K/G238S substitutions propagates beyond the active site to the C-terminus. However our primary observation is that E104K and G238S result in restricted motion of the Ω-loop, in agreement with our models, and we hypothesize that this change in conformational heterogeneity underlies the observed change in substrate specificity.

**Populations of hidden states predict novel cefotaximases.** To definitively test the importance of Ω-loop heterogeneity, we chose new variants designed to similarly restrict the Ω-loop rearrangements, computationally checked that they populate the cefotaximase states and experimentally measured their cefotaximase activities. To the best of our knowledge, none of these variants have been observed in nature, and only one (E104A) has been observed in directed evolution studies[3]. If the electrostatic interactions between residue 104 and the acidic Ω-loop are a key determinant of the populations of the Ω-loop's hidden states, then we reasoned E104D should mimic wild type, whereas E104R should more closely resemble E104K. We also tested the contribution of hydrophobic surface area by substituting aliphatic residues at position 104, predicting that longer side chains would form stronger interactions with the Ω-loop and generate greater cefotaximase activity. In isolation many substitutions at position 104 have only a modest effect on activity[51] and saturation mutagenesis at position 104 suggests no single substitution dramatically alters cefotaxime activity[52], so we also tested all variants in combination with G238S to better assess their impact.

We first constructed MSMs for our designed variants and then assessed the degree to which they populate the cefotaximase states. We then experimentally measured their *in vitro* activities against cefotaxime and the extent to which they confer cefotaxime resistance to *E. coli* (Table 1). Given that similar trends exist in the single and double mutants, we focus on the variants containing the sensitizing G238S mutation. As predicted, E104D/G238S has a similar probability of adopting cefotaximase states as G238S alone and also has similar activity against cefotaxime. E104R/G238S, E104M/G238S and E104A/G238S all populate the cefotaximase states more than G238S alone, as predicted. However, in contrast with expectations based on

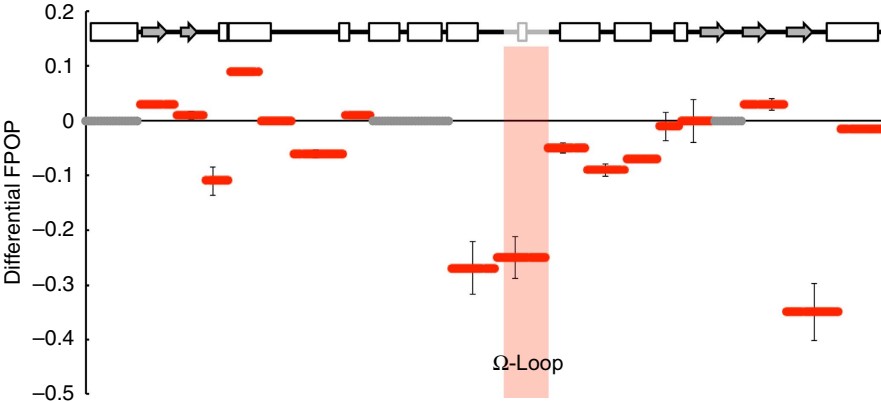

**Figure 5 | FPOP data reveal restricted motion in cefotaximase variant.** Hydroxyl labelling of TEM variants are shown as the difference in percent labelled between M182T/E104K/G238S and M182T to highlight restricted motion in the Ω-loop of the cefotaximase variant. In the secondary structure cartoon the rectangles represent helices, grey shaded arrows represent β-sheets and lines represent loops. The Ω-loop (164–179) is shaded grey on the cartoon and pink in the graph. Grey bars appear where no data was collected, and error bars represent ±1 s.e.m.

charge arguments alone, E104R/G238S does not populate these states as extensively as E104K/G238S (Fig. 2c). These results are consistent with both *in vitro* and *in vivo* experiments, which show E104R/G238S degrades cefotaxime better than G238S but not as well as E104K/G238S (Table 1). Comparing the conformations adopted by E104R/G238S to those of E104K/G238S reveals that Arg104 interacts more strongly with residues 170 and 171 in the Ω-loop, displacing key interactions with the catalytic water[42] in the active site and reducing activity against both benzylpenicillin and cefotaxime (Supplementary Fig. 3). Consistent with our hypothesis that van der Waals interactions play an important role in pinning position 104 to the Ω-loop, E104M confers greater cefotaximase activity than does E104A. In fact, in the wild-type background E104M has greater activity than either of the positively charged variants. Positive epistasis between G238S and E104K, however, results in this double mutant surpassing all others in cefotaximase efficiency. Taken together, these variants imply that both charge and hydrophobic surface area contribute to rate enhancement.

Our approach also tends to successfully identify variants that do not have significant cefotaxime activity. Based on our intuition about the importance of electrostatic interactions, we designed three additional variants intended to favour the closed Ω-loop conformations important for cefotaximase activity. For example, we reasoned that introducing negatively-charged residues behind the acidic Ω-loop at position 164 might increase cefotaxime activity by favouring closed conformations through electrostatic repulsion. However, the MSM for R164E/G238S predicts this variant does not populate the cefotaximase states as much as other known or predicted cefotaximases, and these predictions are consistent with both *in vitro* and *in vivo* experiments (Table 1). R164D/G238S is predicted to populate cefotaximase states more than R164E/G238S but less than most cefotaximases, which is consistent with its measured activity. We also tried changing position 240 to a positively-charged residue with the E104K/E240K variant in an attempt to favour closed Ω-loop conformations through electrostatic attraction. In this case, our MSM suggested this variant could have activity against cefotaxime, but this prediction was not borne out experimentally (Table 1). Despite this exception, the general agreement between our models and experiments demonstrates the added power of MSMs over biochemical intuition alone.

Impressively, the populations of our hidden cefotaximase states provide the most accurate predictions of cefotaxime activity (Fig. 2c, $R = 0.79 \pm 0.03$, compared with $R = 0.30 \pm 0.10$ for

Boltzmann docking and $R = -0.37 \pm 0.07$ for docking against single structures). Furthermore, the correlation is robust to exclusion of the variant used to define cefotaximase states, E104K/G238S ($R = 0.74 \pm 0.04$). This result supports our conclusion that MSMs provide a reasonably accurate depiction of TEM β-lactamases' structural ensembles. Boltzmann docking performed better than docking against single structures by accounting for conformational heterogeneity, but any docking approach is limited by using substrate binding as a proxy for activity. Enzymes undergo conformational changes throughout a catalytic cycle in concert with chemical transformations to the substrate, so these states would not necessarily score well. MSMs, on the other hand, may capture these catalytically-relevant states. Therefore, a powerful approach to classify the activities of new variants is to compare MSMs of their structural ensembles to MSMs for variants with known functions.

## Discussion

Our results demonstrate that accounting for proteins' conformational heterogeneity dramatically improves the predictive power of molecular modelling with common force fields. For example, Boltzmann docking dramatically outperforms docking against single structures by accounting for proteins' hidden states. We anticipate this method will be valuable for predicting resistance to new compounds, especially when clinical variants have not yet been identified. In cases where a set of variants with different activities are known, MSMs can shed light on which conformational states are relevant for different functions. Based on these insights, MSMs can predict the activities of new variants even more accurately than Boltzmann docking by quantifying the populations of hidden states and assessing which of the known variants populate these states most similarly. Both of these approaches will be of great utility for other drug and protein design applications.

## Methods

**Molecular dynamics simulations.** Five 500 ns simulations were run for each variant with Gromacs 4.6.5 (ref. 53) and the Amber03 force field[54] using previously reported settings[12,13], which are reviewed below. Modeller[55] was used to create a homology model of each variant based on PDB 1BTL[10] that was then used as the starting point for simulations. Each of these starting structures was solvated with TIP3P water[56] in a dodecahedron box that extended one nm beyond the protein in any dimension and sodium ions were added to neutralize the charge. This system was energy minimized with the steepest descent algorithm until the maximum force fell below $1,000 \, \text{kJ} \, \text{mol}^{-1} \, \text{min}^{-1}$ using a step size of 0.01 nm and a cut-off distance of 1.2 nm for the neighbour list, Coulomb interactions and

van der Waals interactions. The solvent was then equilibrated in a one ns simulation with a position restraint on all protein heavy atoms (spring constant $1,000 \, \text{kJ mol}^{-1} \, \text{nm}^{-2}$). A long-range dispersion correction was employed for both energy and pressure. All bonds were constrained with the LINCS algorithm[57] and virtual sites[58] were used for all hydrogens to allow a 4 fs time step. Cut-offs of 1.1, 0.9 and 0.9 nm were used for the neighbour list, Coulomb interactions, and Van der Waals interactions, respectively. The Verlet cut-off scheme was used for the neighbour list and particle mesh Ewald[59] was employed for the electrostatics (with a grid spacing of 0.12 nm, PME order 4, and tolerance of 1e − 5). The v-rescale thermostat[60] (with a time constant of 0.1 ps) was used to hold the temperature at 300 K and the Berendsen barostat[61] was used to bring the system to 1 bar pressure. For the production runs, the position restraint was removed and the Parrinello-Rahman barostat[62] was employed. Snapshots were stored every 10 ps. Structures were drawn with PyMOL[63].

**MSM construction and analysis.** MSMs were constructed with MSMBuilder (v2.8)[28,29]. MSMs for individual variants that were used for Boltzmann docking were created by clustering the data for an individual variant with a hybrid clustering method. First, we used a k-centres algorithm based on the r.m.s.d. between heavy atoms in residues surrounding the active site (residues 69–73, 103, 105, 130–132, 165–173, 216, 234–237, and 244) until every cluster had a radius—that is, maximum distance between any data point in the cluster and the cluster centre—< 1.0 Å. Then, three sweeps of a k-medoids update step were used to centre the clusters on the densest regions of conformational space. This procedure resulted in the following number of clusters for each variant: 2,046 for wild type, 1,891 for G238S, 1,693 for E104K, 1,280 for E104K/G238S, 1,467 for E104R, 942 for E104R/G238S, 1,338 for E104A, 1,206 for E104A/G238S, 1,530 for E104D, 2,525 for E104D/G238S, 1,090 for E104M, 780 for E104M/G238S, 1,358 for E240K/E104K, 2,113 for R164D/G238S, and 2,812 for R164E/G238S. We selected the cluster centres for each state as representative structures. We used the representative structures for each cluster as the basis for our Boltzmann docking approach. Another alternative would be to coarse-grain the model by merging clusters into macrostates. However, doing so could easily merge geometrically distinct conformations and lead to inaccurate estimates of the total probability of binding-competent conformations. Supplementary Fig. 4 shows these models satisfy the Markov assumption for lag times as small as 1 ns. Consistent with past work demonstrating that thermodynamics converge far more quickly than kinetics[64], the thermodynamics of our models are insensitive to varying the lag time from 10 ps to 10 ns, so equilibrium populations of each state were determined by calculating a matrix of transition probabilities between every pair of states with the transpose method and a lag time of 10 ps and solving for the normalized left eigenvector of this matrix.

MSMs for comparing the structural preferences of different variants were constructed based on the same set of active-site residues. First, every 100th data point from simulations of each variant were pooled together and clustered into 1,000 states with a k-medoids algorithm. Then the equilibrium probability of each state for a given variant was calculated using the same approach described before using just the data for that variant. Using a common set of states to describe the thermodynamics and kinetics of each variant provides a basis for directly comparing the probabilities that different variants will adopt a given conformation. Of the 1,000 states in this combined state space, we found that 350 of these states have higher populations in E104K/G238S than wild type. As explained in the text, we designate these states as cefotaximase states and employ the sum of the equilibrium populations of these states (with error bars from bootstrapping, as explain below) as a surrogate for cefotaxime activity. Another 399 states are more populated by wild type than E104K/G238S, which we designate as non-cefotaximase states.

Error bars on the population of a subset of MSM states were obtained via bootstrapping. That is, we drew 100 independent subsamples of the data (with replacement) and reported the mean and standard deviation of the sum of the equilibrium probabilities of the subset of states of interest. Correlation coefficients between the populations of subsets of MSM states and experiments were obtained by calculating the Pearson correlation coefficient between all n possible subsets of n − 1 data points (where n is the number of TEM variants) and reporting the mean and standard deviation of these correlation coefficients. Implied timescales were also obtained by calculating the implied timescales of all n possible subsets of n − 1 trajectories (where n is the number of independent simulations) and reporting the mean and standard deviation of these timescales.

Inter-atomic distances were calculated with MDTraj[65]. Two atoms were assumed to be in contact with one another if their centres were within 4 Å. The probability of a contact was calculated by identifying all the states where a pair of residues is in contact and then summing up the equilibrium populations of these states.

**Docking.** Docking against individual structures was performed with Surflex-dock[66]. A TEM-1 crystal structure (PDB 1BTL[10]) was used for wild type, and Modeller[55] was used to create a homology model for each of the other variants based on this crystal structure. The structures of benzylpenicillin and cefotaxime were generated using the Concord module of SYBYL-X 2.1.1 and minimized using the Tripos force field. Surflex-Dock receptor protomols were generated with a threshold of 0.5 and a bloat of 3.0. These protomols were then used to screen various ligands for receptor

complementarity. The Hammerhead scoring function[67] inherent to Surflex was used to score the resulting poses. The default '-pgeom' docking accuracy parameter set was implemented. We also repeated this experiment using a structure with the G238S substitution, PDB 1HTZ[7], as the template for all variants containing the G238S substitution to ensure that subtle differences between 1HTZ and 1BTL do not make a significant difference in the results.

Boltzmann docking was performed using the same settings to dock the substrates against the cluster centres from each state of the MSMs built for an individual variant. The final score was then calculated as the weighted-average of the scores for each state, using the equilibrium probabilities of each state as their weights. Ensemble docking was conducted by taking the same set of scores against cluster centres and ranking compounds based on their highest score against any of the protein structures.

Correlation coefficients between different docking protocols and experiments were calculated in the same manner as the correlations between the populations of subsets of MSM states and experiments, except that we compare to the log of the experimental enzyme efficiencies to put these measurements on an energy scale, like the docking score.

**Protein expression and purification.** TEM-1 was subcloned using NdeI and XhoI restriction sites into the multiple cloning site of a pET24 vector (Life Technologies), and its native export signal sequence was replaced by the OmpA signal sequence to maximize export efficiency[68]. Site-specific variants were constructed via site-directed mutagenesis and verified by DNA sequencing. Plasmids were then transformed into BL21(DE3) Gold cells (Agilent Technologies) for expression under T7 promoter control.

Cells were induced with 1 mM IPTG at OD = 0.6 and grown at 18 °C for 15 h before harvesting. TEM β-lactamases were isolated from the periplasmic fraction using osmotic shock lysis: Cells were resuspended in 30 mM Tris pH 8, 20% sucrose and stirred for 10 min at room temperature. After centrifugation, the pellet was resuspended in ice-cold 5 mM MgSO$_4$ and stirred for 10 min at 4 °C. After centrifugation, the supernatant was dialyzed against 20 mM sodium acetate, pH 5.5 and purified using cation exchange chromatography (BioRad UNOsphere Rapid S column) followed size exclusion chromatography (BioRad ENrich SEC 70 column) into storage buffer (20 mM Tris, pH 8.0).

**Fast photochemical oxidation of proteins.** TEM variants were submitted to FPOP by irradiating with an excimer laser pulse on the flowing mixture of protein and H$_2$O$_2$, as described originally[69]. Briefly, 10 μM of protein in PBS, pH 7.4, was mixed with the 20 mM L-glutamine and 15 mM H$_2$O$_2$ at room temperature and flowed in a 150 μm i.d. capillary by a syringe pump (Harvard Apparatus, Holliston, MA, USA) at a rate of 24 μl min$^{-1}$ past a transparent laser window where the flowing solution was irradiated by 248 nm KrF excimer laser (GAM Laser Inc., Orlando, FL, USA). The laser power was adjusted to ∼35 mJ pulse$^{-1}$ at a frequency of 7 Hz, ensuring an exclusion volume fraction of 20% having no laser irradiation. The laser-irradiated sample was collected in a vial containing 500 nM catalase and 70 mM L-methionine to decompose excess H$_2$O$_2$. The sample was frozen in liquid nitrogen and then stored in − 80 °C until further analysis.

For each mutant the labelling reaction was done in triplicate. The sample was divided into two parts. One part was analysed using a Bruker Maxis Q-TOF mass spectrometer (Billerica, MA, USA) under denaturing conditions to check for hydroxyl radical labelling at the protein level. The other was treated with 20 mM TCEP, PBS, pH 7.4 at 45 °C for 1 h to reduce the disulfide bond in PBS pH 8.0. The alkylated protein sample was immediately purified using the 2-D clean-up kit (GE Healthcare, NJ, USA), denatured in presence of 8 M urea and digested overnight with trypsin in 100 mM TEAB, pH 7.8 at 37 °C. The resulting peptides were separated by reversed-phase HPLC using a Dionex Ultimate 3,000 RSLCnano system (Waltham, MA, USA) and introduced to a Thermo QExactive Plus mass spectrometer (Waltham, MA, USA) via nano-ESI. Peptide ions were fragmented in a data-dependent mode in which the 10 most abundant components were selected by the spectrometer for fragmentation. Unmodified and modified peptides were identified by Mascot (Matrix Science, Boston, MA, USA) and Byonic (Protein Metrics, San Carlos, CA, USA) and confirmed by manual inspection.

**Activity measurements.** Enzyme activities against benzylpenicillin (BP) and cefotaxime (CFX) substrates were measured in 50 mM potassium phosphate, 10% glycerol (v:v) pH 7.0 at 25 °C using 2, 10 or 200 nM enzyme. The reaction was monitored at 232 nm ($\varepsilon_{BP} = -1,096 \, M^{-1} \, cm^{-1}$) or 265 ($\varepsilon_{CFX} = -6,643 \, M^{-1} \, cm^{-1}$) using a Cary 100 UV–vis spectrophotometer (Agilent Technologies). Velocities were plotted as a function of substrate concentration and fit by the Michaelis–Menten equation to extract $k_{cat}$ and $K_m$ values. Enzymes that did not exhibit saturation behaviour under the tested conditions were fit by a line, and the slopes are reported as $k_{cat}/K_m$.

**MIC measurements.** Site-specific variants of TEM-1 were constructed via site-directed mutagenesis of the pBR322 plasmid and verified by DNA sequencing. Plasmids were then transformed into BL21(DE3) cells (Intact Genomics) and DH5α cells (Life Technologies) to create strains in which β-lactamases are expressed using a native promoter.

Antibiotic resistance of the strains was determined by measuring their minimum inhibitory concentrations (MIC90's) using the broth microdilution method according to the Clinical and Laboratory Standards Institute (CLSI, formerly the NCCLS) guidelines[70]. Each well of a 96-well microtiter plate was filled with 75 μl of sterile Mueller Hinton II (MHII) media broth (Sigma). Each antibiotic was dissolved in water making a 20 mM solution, then diluted with sterile MHII media broth to 192 mM (BP) or 288 μM (CFX). Exactly 75 μl of the compound solution was added to the first well of the microtiter plate, and twofold serial dilutions were made down each row of the plate. Exactly 75 μl of bacterial inoculum ($5 \times 10^5$ c.f.u. ml$^{-1}$) was then added to each well giving a total volume of 150 μl per well and compound concentration gradients of 48 mM–23 μM (BP) and 72–0.04 μM (CFX). The plate was incubated at 37 °C for 17 h, and then each well was examined for bacterial growth. The MIC90 was recorded as the lowest compound concentration required to inhibit 90% of bacterial growth as judged by turbidity of the culture media relative to a row of wells filled with a water standard. Gentamicin was included in a control row at a concentration gradient of 174–0.09 μM.

**Data availability.** The experimental data supporting the findings of this study as well as the computer code and simulations are available from the corresponding author upon request.

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

## Acknowledgements

We are grateful to Gaya Amarasinghe, Gautam Dantas and Rohit Pappu for helpful comments. G.R.B. holds a Career Award at the Scientific Interface from the Burroughs Wellcome Fund. The mass spectrometry was supported by the NIH (Grant # P41GM103422 to M.L.G). We are grateful to NVIDIA Corporation for the GTX Titan X used to run preliminary simulations. This work was also funded by NSF CAREER Award MCB-1552471 to G.R.B.

## Author contributions

K.M.H, C.M.W.H., S.D, M.G. and G.R.B. designed and performed research; K.M.H., C.M.W.H., S.D, M.G. and G.R.B. analysed data and wrote the paper.

## Additional information

**Competing financial interests:** The authors declare no competing financial interests.



