## [Peer Review File · Nature Communications]

Reviewers' Comments:

Reviewer #1 (Remarks to the Author)

Reviewer: James Fraser (with Daniel Keedy)

This manuscript focuses on understanding how resistance mutations in TEM-1 beta-lactamase convey novel substrate specificities to more efficiently degrade new drug molecules. The focus is on using molecular dynamics (MD) simulations and Markov State Models (MSMs) to represent multiple conformational states of the protein that may be differentially populated in different mutants and contribute to different substrate specificities. The question is old, but the methods are new. Their computational predictions are validated with a mass spectrometric labeling technique to confirm the existence of multiple states (albeit not their actual structures), and further compared against in vitro and in vivo activity assays.

The quality of the computational work and the experimental tests of flexibility is very high - but the link to the "mystery" of the mutations is on shakier ground. The authors have not given enough of a look to previous experimental work in the area - which compounded with some questionable choices in which template structures to use for homology models - creates a bit of a false pretense for some of the motivation of the work. cursory analysis of the previous structures indicates structural changes that are not properly highlighted by Fig 1 or the indication of an RMSD. Why are homology models based on WT, not the mutations + the M182 variant, used for docking? The conformational changes in the omega loop and G238 loop are surely the driving forces behind the bad predictions. These changes are not present in the template, but some are present in the G238S/E104K/M182T structure (1HTZ) or other structures containing G238S in other backgrounds (eg. 4OP8, 4OPQ, 4OPR, 4OPZ, 4OQ0, 4OQI)! The paper would benefit from providing structural detail (more than just the inset of Fig 3) of what the new states look like and how they compare to the large number of existing crystal structures out there.

That said, there are are two main contributions:

- (1) Single-structure docking is extended to "Boltzmann docking" against the cluster centers from the MSM, which improves the correlation of docking score with activity. This method is potentially quite general and goes far beyond previous flexible docking approaches (incl. ours with Shoichet) - this is very nicely done!!!
- (2) Particular "hidden" conformations (not present in the crystal structures - although I note that the G238S state is probably there in most of the crystal structures that contain this mutation - in combination with others!) are identified as being more populated in the mutant vs. the wildtype; the calculated populations of these conformations are shown to correlate more robustly with activity. This method yields a better improvement in correlation of docking score with activity, but is not as general since it requires pre-knowledge of particular natural/clinical mutations. Accounting for hidden conformations leads to significantly better predictions of enzyme activity is a notable contribution - even

with the caveats that binding and catalysis will not in general precisely match, and that interactions of the protein and ligand will mutually impact both conformational ensembles and therefore predictions of binding and activity (which the authors note as an exciting future direction).

Other major points:

The observation that a rigidified omega-loop helps catalysis of cefotaxime is very cool - however, I'm still unclear **why** that is true. The authors should provide some more insight / intuition into the structural basis of this point. Do your Boltzmann-docked models reveal that, for example, cefotaxime can't sterically fit into the WT MSM models (in which the loop is more flexible) but can fit well into the E104K/G238S MSM models (in which the loop is more rigid - potentially in a way that accommodates the ligand well)? This would confirm the classic hypothesis from the Stevens directed evolution/resistance NSB paper on these mutants.

The structural figures are completely confusing! For example: The Figure 1 inset is confusing - there are 2 models, so why does there appear to be 3 conformations for residue 100? The answer is that the E104K mutation actually does cause a significant **local** conformational change: the sidechain moves way more than the 0.35 Å number the authors quote as the degree of conformational change. (BTW, is that number C-alpha RMSD or all-heavy-atom or ...?)

Obviously, the actual activities of these variants for different substrates are in reality a complex interplay between the precise positioning of chemical groups in individual structures, **and** the frequency with which those structures are accessed within the larger ensemble. It would be nice to get that point across, while still highlighting the unique contributions being made here. We tried to do it in the paper with Tawfik on this exact system, but fell short. Because of the increased sampling and potential for true Boltzmann probabilities from the MS, this has a better shot of working.

Minor scientific points:

It would be nice to provide models of the cefotaximase states and the penicillinase states as supplementary material.

How do the kinetic rates of conformational interconversion of the omega-loop from the MSM fits compare to the microsecond experimental timescale of the FPOP experiment? The latter should be faster for these results to be meaningful. Fig. S4 may relate to this point, but the exact relationship between a "Relaxation timescales for TEM-1" (i.e. the whole protein), as indicated in that figure's caption, and the motional timescale for the particular loop of interest, is not immediately obvious.

More careful explanations of the template structures and homology models are needed in the methods

Typos:

Abstract:"ensemble-nature" no hyphen

Figure 1: Neither model is green - you mean blue

117-118: "Boltzmann docking of cefotaxime against each of our variants correlates well with our 118 experiments (Fig. 2b)" - Did you measure the enzyme kinetics?

158-159: determinants -> plural

Reviewer #2 (Remarks to the Author)

A. This manuscript reports a novel computational method, essentially the application of Markov state models to docking, for improved functional predictions from ensemble modelling. The computational work is accompanied by FPOP assay for omega loop dynamics and cefotaximase activity assays to test a set of rationally designed mutants of TEM-1. The work is interesting - at least from a computational perspective - and the manuscript is well written.

B. The 'Boltzmann docking' method that is being introduced in this work could be of substantial interest to the community, as it is a novel approach for ensemble-based docking predictions. Having said that, the authors do overplay the novelty in a number of cases. For instance, the authors compare their ensemble predictions to 'single structure' predictions, which is a reasonable start, but it would be far better to test their method against *other ensemble-based methods*, a number of which have been around for quite some time (Proteins. 2007 Feb 1;66(2):399-421 ; J. Chem. Inf. Model., 2014, 54 (7), pp 2127-2138... etc.).

On the Biological side, the novelty is also not quite what is suggested by the authors. For one thing, the 'hidden' conformations are indeed 'hidden' in conventional crystallography, but can be detected and characterized by biophysical NMR (particularly CPMG relaxation dispersion: Nature 438, 117-121), H/D exchange (particularly millisecond H/D exchange for loops: Lab Chip, 2013,13, 2528-2532) and even FPOP in some cases. This isn't to say that it's unhelpful to have computational methods that predict these conformations from crystal structures, but the fact remains that these 'hidden' conformations are not in fact invisible to experimentalists. Directly measuring changes in omega loop dynamics in various omega loop mutants, for instance, would be fairly straightforward experimentally, and would lead presumably to the same biological conclusion that is presented in this work. (Restricted omega loop dynamics = higher activity)

The authors also claim that 'crystal structures of TEM variants do not provide a clear explanation of their functional parameters'. In fact, for the mutants being studied, there seems to be a consensus, based on crystallographic data that cephalosporinase activity-enhancing mutations act by 'opening' the active site (this is partly acknowledged by the authors), which strikes me as being how a crystallographer would describe the 'pinning' effect that is reported in the current work. Having said this, the computational model does provide additional indications as to the mechanism by which the active site is 'opened'.

C. The data are generally of high quality. There is some question of the appropriateness of the FPOP method for measuring loop dynamics, since it is seen largely as being sensitive to solvent accessibility, but it is possible that 'pinning' effect would restrict solvent access to the loop somewhat.

D. The authors did test their hypothesis against a significant and sufficient number of mutants in my view. However it would have been nice to see more *sites* since this entire region is known to undergo mutations that affect specificity: J Bacteriol. 1996 Apr;178(7):1821-8. Moreover, many of those mutations are non-specific, suggesting that simply increasing the dynamics of the omega loop (as opposed to pinning it) also causes an increase in cephalosporinase activity.

E. The main conclusion - that ensemble-based docking can provide enhanced functional predictions and insights for enzymes is mostly supported by the data (although, it should be noted that there are many enzyme systems for which the 'hidden' intermediates are more directly linked to the catalytic reaction coordinate than substrate binding. The TEM-1 / Omega-loop system happens to be ideal for 'catalytic efficiency' effects that are mostly derived from substrate binding enhancement)

F. Only suggestion would be a more thorough investigation of different mutation sites in the omega loop, but I do believe the number of mutants currently looked at is sufficient.

G. The references are largely adequate, but should include some discussion and references to previous work in ensemble-based docking. Wouldn't hurt to cite experimental methods for detecting and characterizing 'hidden' conformations as well.

H. The clarity of writing was good.

Reviewer #3 (Remarks to the Author)

In this manuscript, the authors nicely combined MSMs with biochemical and mass spectrometry footprinting experiments to successfully elucidate how different variants of TEM β -lactamase (TEM) can lead to their dramatically functional differences. This cannot be explained by static crystal structures. The MSMs constructed from MD simulations not only facilitated the Boltzmann docking which outperforms the single-structure docking, but also identified the unprecedented hidden states of TEM responsible for hydrolyzing cefotaxime. The flexibility of Ω -loop predicted by MSMs was confirmed by mass spectrometry footprinting. Furthermore, the author used MSMs together with measurements of enzyme activity and MIC to show that both electrostatic interactions and hydrophobic surface area could have led to stronger enzyme activity.

This work holds great promise to facilitate the development of new drugs to fight against antibiotic resistance. Using TEM as an example, the authors shed new light on the structural and molecular basis of cefotaxime resistance. The methodology (MSM-guided mass spectrometry footprinting and functional assays) can serve as a powerful means for investigating structural basis of other enzyme functions. That being said, I also have a few questions and suggestions concerning the details of this manuscript. Therefore, I would like to recommend its publication at Nature Communications after the following comments are addressed:

1. The authors showed that a few hidden states (e.g. in Fig. 2C) are important for predicting cefotaxime activity. I am curious how did the authors select these important hidden states from their

MSM containing 1000 states? I would also suggest the authors to display and discuss representative structures from these hidden states.

2. This is a related question on the hidden states. In Fig. 2C, the authors show strong correlations between the populations of hidden states with the experimental catalytic rates. If the Boltzmann weighted docking scores to these hidden state conformations rather than their populations are displayed, I am wondering if the same or even better correlation will hold? Anyway, I would suggest that authors to include some discussions on docking to these hidden states.

3. In the Boltzmann docking, docking scores from all MSM states are weighted averaged. While in typical ensemble docking studies, only top docking poses are selected, while others are discarded. If the initial binding to one particular can be significantly stabilized by subsequent induced fit, I would think the ensemble docking approach might work better. Could the authors add in some comment on the pros and cons of these two different approaches?

4. The Markovian lag time of their MSM is at over 1ns, while the authors use 10ps as the lag time to construct their MSM to calculate equilibrium populations. I agree with the authors that thermodynamics generally reaches convergences faster than kinetics. To make it more convincing, I would suggest the authors to calculate the equilibrium population with different lag times (e.g. 20ps) and show the invariance of obtained equilibrium populations.

5. The following paper discussed an approach of using MSM followed by protein-RNA docking to elucidate the miRNA-Ago recognition mechanisms, which could be relevant to the current work when the authors work on further elucidating the molecular recognition mechanisms:

Jiang, H., Sheong, F.K., Zhu, L., Gao, X., Bernauer, J., Huang, X., PLoS. Comp. Bio., 11(7): e1004404, (2015)

Gu, S., Silva, D.A., Meng, L., Yue, A., Huang, X., PLoS. Comp. Bio., 10(8):e1003767, (2014)

6. The plots in Figure 2.b shows that Boltzmann docking outperforms the single-structure docking in predicting the specificity of TEM variants. It would be helpful if the docked structures accompany these plots to show the structural details of TEM-cefotaxime interactions.

7. It seems that the correlation coefficients (R) for Supplementary Figure 1. a&b are missing.

Reviewer #4 (Remarks to the Author)

The manuscript "Modeling proteins' hidden conformations to predict antibiotic resistance" describes a combined computational-experimental study of the role of conformational heterogeneity stabilized by mutations in determining TEM beta-lactamase (TEM) specificity for beta-lactam antibiotics. This is surprising, considering the traditional viewpoint that TEM is a rather rigid protein. Despite this, the ability of TEM to access alternative conformations can explain how mutations far from the substrate binding site can alter substrate specificity by up to several orders of magnitude.

While there are some weak statistical components of the manuscript---namely, a lack of statistical rigor in correlation coefficients, and claims made about Boltzmann docking that are almost certainly artifacts of the poor statistical power of $N = 12$ data points---the overall story is extremely interesting, and would be of great interest to the readership of Nature Communications. Taken as a whole, the characterization of hidden states by molecular simulation, the footprinting experiments, and the engineered mutants do seem to suggest a significant role for these "hidden states" in catalysis.

I recommend publication after fixing the statistical deficiencies and other issues noted below.

Issues that should be addressed before the manuscript is acceptable for publication:

- * All correlation coefficients R should incorporate some form of uncertainty estimate arising from the use of a finite dataset.

Otherwise, there is no way to evaluate whether statements such as " $R = 0.35$ versus $R = 0.07$ " carry any statistical significance.

At minimum, this should be bootstrapped statistical errors, or better yet, confidence intervals.

The authors are referred to <http://dx.doi.org/10.1007%2Fs10822-014-9753-z> for a more detailed description of the appropriate statistics for R in molecular modeling.

- * The quantities being correlated (line 94) are not clearly stated.

- * The discussion attributing failures of docking solely to inadequacies in the forcefield (lines 93-103) is unfairly shortsighted. Both the fact that docking scores a single structure (rather than computing the free energy of binding by integrating over conformational space) and the neglect of protonation/tautomeric states could also be contributors, as could deficiencies in sampling within the conformations (such as sidechain conformational sampling). Even the manner in which the protein structure was prepared for docking could contribute. This experiment is really only useful in comparison with the next experiment in which everything else is kept constant except weighted contributions from multiple conformational states are included.

- * The manner in which the "representative structure" from each MSM state was selected---and the manner in which the MSM states were selected---should be stated. Are these microstates or lumped macrostates? How many were there to choose from?

- * I count 12 measurements being depicted in Figure 2b. Using Eq. 54 of [<http://dx.doi.org/10.1007%2Fs10822-014-9753-z>] to compute the minimum R value outside of the 95% confidence interval for no correlation gives a minimum R threshold of 0.57. That means that, with only $N = 12$ data points, the 95% confidence interval for R for uncorrelated data ($R = 0$) includes all R values up to 0.57. Therefore, there is insufficient data to state that the improved correlation achieved by Boltzmann docking ($R = 0.35$ vs $R = 0.07$) is due to anything but chance.

* "We queried the MSMs for states that are significantly more populated by one sequence over the other." Can you clarify whether microstates (which are simply configurational clusters) or lumped macrostates (which might represent kinetically meaningful conformations) are being referred to here?

* Why should the docking score correlate with k_{cat}/K_m and not $\log(k_{cat}/K_m)$ or $\log(K_m)$? The docking score should be on an energy ($\log K_d$) scale, while the measured k_{cat} , K_m , or k_{cat}/K_m are on exponentiated energy scale.

* "populations of our hidden cefotaximase states" - how was this quantity defined? I couldn't seem to find a quantitative definition of how this was computed---was this just microstates with higher populations in WT as opposed to mutant? How robust is that set to statistical error in the microstates?

Interestingly, this correlation coefficient ($R = 0.83$) is large enough to likely be significant!

* Molecular dynamics simulations: This section is lacking sufficient detail for a competent practitioner to reproduce the simulations.

* "with a position restraint on all 300 protein heavy atoms" - what spring constant was used?

* what temperature and pressure was used?

* was a long-range isotropic dispersion correction used?

* which sites were turned into virtual sites?

* what PME parameters were used (grid spacing, order, error tolerance)?

* what v-scale interval?

* How often were snapshots written?

* MSM construction and analysis:

* Which version of MSMBuilder was used?

* How many resulting microstates were there?

* Was any lumping performed, or were the microstates used as "states" throughout?

* Supplementary Figure 4: This plot is lacking error bars, and the x- and y-axes are both lacking units.

* Supplementary Table 1: I'm not sure this table satisfies the NPG Statistical Guidelines [<http://www.nature.com/srep/publish/guidelines>] in its current form.

Reviewer #1 (Remarks to the Author):

Reviewer: James Fraser (with Daniel Keedy)

This manuscript focuses on understanding how resistance mutations in TEM-1 beta-lactamase convey novel substrate specificities to more efficiently degrade new drug molecules. The focus is on using molecular dynamics (MD) simulations and Markov State Models (MSMs) to represent multiple conformational states of the protein that may be differentially populated in different mutants and contribute to different substrate specificities. The question is old, but the methods are new. Their computational predictions are validated with a mass spectrometric labeling technique to confirm the existence of multiple states (albeit not their actual structures), and further compared against in vitro and in vivo activity assays.

The quality of the computational work and the experimental tests of flexibility is very high - but the link to the "mystery" of the mutations is on shakier ground. The authors have not given enough of a look to previous experimental work in the area - which compounded with some questionable choices in which template structures to use for homology models - creates a bit of a false pretense for some of the motivation of the work. cursory analysis of the previous structures indicates structural changes that are not properly highlighted by Fig 1 or the indication of an RMSD. Why are homology models based on WT, not the mutations + the M182 variant, used for docking? The conformational changes in the omega loop and G238 loop are surely the driving forces behind the bad predictions. These changes are not present in the template, but some are present in the G238S/E104K/M182T structure (1HTZ) or other structures containing G238S in other backgrounds (eg. 4OP8, 4OPQ, 4OPR, 4OPZ, 4OQ0, 4OQI)! The paper would benefit from providing structural detail (more than just the inset of Fig 3) of what the new states look like and how they compare to the large number of existing crystal structures out there.

Response: We made significant changes to the main text and figures to incorporate more of the previous experimental work on TEM. Specifically, we add a paragraph (below) contrasting our structural model with the existing model (ie that movement of the 238-loop results in an more open active site, which is required to bind large substrates like cefotaxime). Briefly, our model suggests that the active site is more open in wild-type than in the cefotaximase variants, and that this openness is due to movement of the omega-loop. The most populated cefotaximase states captured by our MSM do not have the structural rearrangements in the 238-loop observed in the crystal structures you mention.

"Both of these substitutions have been studied extensively, but to our knowledge we are the first to present a model linking either mutation to restricted motion in the Ω -loop. Crystal structures of G238S-containing variants present conflicting models for how the 238-loop and Ω -loop interact. The TEM-52 structure lacks hydrogen bonds between the two loops⁷, but other structures capturing multiple conformations of the Ser238 side chain show it can hydrogen bond to the backbone carbonyls of Asn170 or Glu171.¹¹ We also observe formation of a hydrogen bond between Ser238 side chain and N170 backbone, but the precise conformation of the 238-loop differs. The crystal structures show that the 238-loop moves 2-4 Å away from the Ω -loop (measured by the C $_{\alpha}$ positions of Glu240 in G238S-containing structures versus TEM-1), which results in a small widening of the active site. In contrast, we observe that both loops assume conformations more similar to the wild-type crystal structure in our most populated cefotaximase states (Fig. 4). Our model suggests that rather than opening up the active site, G238S and E104K maintain a closed active-site architecture by pinning down the Ω -loop. In the absence of these substitutions, the Ω -loop exhibits greater flexibility and accesses conformations that increase its solvent exposure. This opening event in wild-type is a more dramatic change than the small widening caused by movement of the 238-loop in G238S-containing crystal structures. The models are not necessarily mutually exclusive, but ours challenges the proposal that wild-type cannot degrade cefotaxime because it is simply too big to fit in the active site. Our Boltzmann docking analysis supports our model because the structures with the highest docking score

against cefotaxime are similar between wild-type and E104K/G238S (Fig. 3). Both docked structures have loop conformations resembling the wild-type crystal structure (Fig. 3, insets). We do not observe active-site opening when cefotaxime is bound, and the larger substrate can, indeed, fit in the wild-type active site without steric conflict. It is the fact that this binding-competent state is not highly populated in wild-type that underlies its low activity against cefotaxime. Finally, kinetic analysis of cefotaximase variants shows that higher catalytic efficiencies likely result from lower K_m values. Because acylation is the rate-limiting step, K_m approximates the substrate binding affinity.⁴⁴ Cefotaximase variants also have lower K_m s for benzylpenicillin (Table 1), which suggests restricting motion in the Ω -loop helps the enzyme bind both substrates. The low k_{cat} values observed for these variants might indicate that Ω -loop flexibility is important for another step in the catalytic cycle, such as product release. We propose that the conformation captured by the wild-type crystal structure, particularly of the Ω -loop, is the binding-competent state for a diverse set of substrates, and that E104K/G238S stabilizes this state to allow for more promiscuous binding.”

We added a new figure (Fig. 3) to show what the new states look like and compare them with existing structures.

Figure 3. Highest ranking poses from Boltzmann docking of cefotaxime against (a) wild-type (green) and (b) E104K/G238S (orange). Both structures are shown overlaid with the crystal structure of TEM-1 (blue, PDB ID 1BTL) for reference. Cefotaxime is in cyan. Insets depict key loops flanking the active site with the catalytic Ser70 shown in red.

Your point about the robustness of our homology models to the starting structure is well taken. Our initial rationale for choosing one starting structure for all the variants was to test the generalizability of the

method for proteins that do not have multiple structures. In the case of TEM, however, we can see how this might seem like an unfair baseline comparison meant to artificially inflate the performance of our other methods. So, we repeated the homology modeling and docking using 1HTZ (E104K/G238S/M182T) for variants containing G238S and found this only makes a slight difference in the correlation ($R = -0.37 \pm 0.07$ for wild-type homology models versus $R = -0.30 \pm 0.05$ for 1HTZ homology models). We interpret this to mean that the crystal structure doesn't provide enough information to discriminate between variants that can or cannot degrade cefotaxime. We added this analysis to the main text:

"If the ability of ground-state structures of TEM variants to bind different substrates specifies their activities, then it should be possible to make predictions by docking substrates against crystal structures, when available, or homology models when structures have not yet been solved. To test this hypothesis, we chose to study a series of variants with differing activities against cefotaxime, a substrate for which new activity evolved in the clinic. First, we built homology models for the variants. Although there are crystal structures for many different variants of TEM, this is not true for other enzymes we might wish to study. To preserve the generalizability of this method, we chose to build all the homology models using TEM-1¹⁰ to mimic situations where only one structure might be available. Then, we docked cefotaxime against the active sites of each of these variants. Because we use substrate docking as a proxy for activity, we expect higher scores to predict greater activity against the docked substrate.

Docking against single structures of each variant completely fails to predict their abilities to degrade cefotaxime, as measured by their k_{cat}/K_m values. In fact, we observe that the docking scores and activities are anticorrelated ($R = -0.37 \pm 0.07$, Fig. 2a). Examining our homology models of variants containing the substitution G238S revealed that they do not capture a subtle shift in the 238-loop (residues 238-242) that has been observed in a number of crystal structures of variants with this substitution. To check that failure to capture this subtle shift is not responsible for the poor correlation in Fig. 2a, we repeated the experiment using a TEM-52 crystal structure⁷ (E104K/G238S/M182T) as the template for all variants containing the G238S substitution. However, this alternative protocol did little to improve the prediction, again showing an anticorrelation between docking and activity ($R = -0.30 \pm 0.05$)."

That said, there are two main contributions:

(1) Single-structure docking is extended to "Boltzmann docking" against the cluster centers from the MSM, which improves the correlation of docking score with activity. This method is potentially quite general and goes far beyond previous flexible docking approaches (incl. ours with Shoichet) - this is very nicely done!!!

(2) Particular "hidden" conformations (not present in the crystal structures - although I note that the G238S state is probably there in most of the crystal structures that contain this mutation - in combination with others!) are identified as being more populated in the mutant vs. the wildtype; the calculated populations of these conformations are shown to correlate more robustly with activity. This method yields a better improvement in correlation of docking score with activity, but is not as general since it requires pre-knowledge of particular natural/clinical mutations. Accounting for hidden conformations leads to significantly better predictions of enzyme activity is a notable contribution - even with the caveats that binding and catalysis will not in general precisely match, and that interactions of the protein and ligand will mutually impact both conformational ensembles and therefore predictions of binding and activity (which the authors note as an exciting future direction).

Other major points:

The observation that a rigidified omega-loop helps catalysis of cefotaxime is very cool - however, I'm still unclear *why* that is true. The authors should provide some more insight / intuition into the structural basis of this point. Do your Boltzmann-docked models reveal that, for example, cefotaxime can't sterically fit into the WT MSM models (in which the loop is more flexible) but can fit well into the E104K/G238S MSM models (in which the loop is more rigid - potentially in a way that accommodates the ligand well)? This would confirm the classic hypothesis from the Stevens directed evolution/resistance NSB paper on these mutants.

Response: We added an explanation for how changes in omega-loop mobility could result in higher cefotaximase

activity in the paragraph cited above. Briefly, G238S lowers K_m values for both benzylpenicillin and cefotaxime, so we believe rigidification of the omega loop acts by increasing binding affinity for both substrates (and maybe any substrate with a beta-lactam core structure). The E104 substitutions do not effect K_m on their own, but in the background of G238S, they also lower K_m . Our model is that rather than stabilizing an alternative state, these substitutions stabilize a state that is catalytically relevant for both substrates. This is at the expense of other catalytically relevant states, possibly product release, as evidenced by the decrease in k_{cat} for benzylpenicillin.

The structural figures are completely confusing! For example: The Figure 1 inset is confusing - there are 2 models, so why does there appear to be 3 conformations for residue 100? The answer is that the E104K mutation actually does cause a significant *local* conformational change: the sidechain moves way more than the 0.35 Å number the authors quote as the degree of conformational change. (BTW, is that number C-alpha RMSD or all-heavy-atom or ...?)

Response: We apologize for the confusing figure and have attempted to clarify it (below). In the inset, we previously showed the active site residues (70-73, 130-132, 166-170, 234-237, 244) and residues 238 and 104 as a way to orient to the blown-up overlay. The “three conformations” comes from confusing Asp131 from the two overlaid structures with Glu104 in the wild-type structure. Residue 104 does differ between the two structures (gamma carbons are 2.9 Å apart), but E104K moves away from the omega-loop, which is the opposite direction of the motion detected by our MSM model in the cefotaximase states. In the new figure, we remove 238 and 104 from the inset for clarity. We hope it is more clear that there are minimal differences in the active site architecture between the variants.

RMSD values account for all heavy atoms. We have clarified this in the figure legend.

Figure 1. Overlay of TEM-1 (blue, PDB 1BTL) and TEM-52 (yellow, PDB 1HTZ) reveals subtle conformational differences (heavy-atom RMSD = 0.60 Å), particularly in the loops containing the mutations, but very similar active-site architectures (inset, heavy-atom RMSD = 0.33 Å). Residues at positions 104 and 238 flank the active site and are shown in pink spheres.

Obviously, the actual activities of these variants for different substrates are in reality a complex interplay between the precise positioning of chemical groups in individual structures, *and* the frequency with which those structures are accessed within the larger ensemble. It would be nice to get that point across, while still highlighting the unique

contributions being made here. We tried to do it in the paper with Tawfik on this exact system, but fell short. Because of the increased sampling and potential for true Boltzmann probabilities from the MS, this has a better shot of working.

Response: Agreed. We added text to clarify that docking uses substrate binding as a surrogate for activity (below). And with this caveat, we ask if including conformational heterogeneity makes a difference in predictive ability. We conclude that it does based on Boltzmann docking outperforming docking against single structures. However, our best prediction comes from the MSM cefotaximase states, which demonstrates that this method may capture a wider array of catalytically relevant states.

“Boltzmann docking performed better than docking against single structures by accounting for conformational heterogeneity, but any docking approach is limited by using substrate binding as a proxy for activity. Enzymes undergo conformational changes throughout a catalytic cycle in concert with chemical transformations to the substrate, so these states would not necessarily score well. MSMs, on the other hand, may capture these catalytically relevant states. Therefore, a powerful approach to classify the activities of new variants is to compare MSMs of their structural ensembles to MSMs for variants with known functions.”

Minor scientific points:

It would be nice to provide models of the cefotaximase states and the penicillinase states as supplementary material.

Response: Good point. We created a new main text figure to show the top cefotaximase and non-cefotaximase states (Fig. 4, below).

Figure 4. A crystal structure of TEM-1 (blue, PDB 1BTL) is overlaid with the two most populated structures taken from MSMs of the (a) non-cefotaximase states (green), which are favored by wild-type and the (b) cefotaximase states (orange), which are favored by the E104K/G238S. (c) Large structural rearrangements in the Ω -loop distinguish low-energy non-cefotaximase states from cefotaximase states, which more closely resemble the conformation captured by the crystallographic structure. Residues 104 and 238 are shown in spheres.

How do the kinetic rates of conformational interconversion of the omega-loop from the MSM fits compare to the microsecond experimental timescale of the FPOP experiment? The latter should be faster for these results to be meaningful. Fig. S4 may relate to this point, but the exact relationship between a "Relaxation timescales for TEM-1" (i.e. the whole protein), as indicated in that figure's caption, and the motional timescale for the particular loop of interest, is not immediately obvious.

Response: The FPOP labeling rate (microseconds) is similar to the timescale of Ω -loop motions in wild-type and faster than Ω -loop motions in E104K/G238S; however we are not interpreting labeling rates, but rather comparing the labeled populations at a given time point. If labeling were not fast enough, it is possible we would not be able to detect differences, even if they were real, but the fact that we see differences between variants indicates some change in solvent accessibility of the loop. We are in the process of measuring rates to determine whether these differences are in the opening rate or equilibrium stability of the loop.

More careful explanations of the template structures and homology models are needed in the methods

Response: We made the following changes to the Methods to clarify:

“Docking against individual structures was performed with Surflex-dock.⁶⁶ A TEM-1 crystal structure (PDB 1BTL¹⁰) was used for wild-type, and Modeller⁵⁵ was used to create a homology model for each of the other variants based on this crystal structure. The structures of benzylpenicillin and cefotaxime were generated using the Concord module of SYBYL-X 2.1.1 and minimized using the Tripos force field. Surflex-Dock receptor protomols were generated with a threshold of 0.5 and a bloat of 3.0. These protomols were then used to screen various ligands for receptor complementarity. The Hammerhead scoring function⁶⁷ inherent to Surflex was used to score the resulting poses. The default ‘-pgeom’ docking accuracy parameter set was implemented. We also repeated this experiment using a structure with the G238S substitution, PDB 1HTZ,⁷ as the template for all variants containing the G238S substitution to ensure that subtle differences between 1HTZ and 1BTL do not make a significant difference in the results.”

Typos:

Abstract: "ensemble-nature" no hyphen

Figure 1: Neither model is green - you mean blue

117-118: "Boltzmann docking of cefotaxime against each of our variants correlates well with our 118 experiments (Fig. 2b)" - Did you measure the enzyme kinetics?

158-159: determinants -> plural

Response: Thank you for your close reading. We made these corrections to the text.

Reviewer #2 (Remarks to the Author):

A. This manuscript reports a novel computational method, essentially the application of Markov state models to docking, for improved functional predictions from ensemble modelling. The computational work is accompanied by FPOP assay for omega loop dynamics and cefotaximase activity assays to test a set of rationally designed mutants of TEM-1. The work is interesting - at least from a computational perspective - and the manuscript is well written.

B. The 'Boltzmann docking' method that is being introduced in this work could be of substantial interest to the community, as it is a novel approach for ensemble-based docking predictions. Having said that, the authors do overplay the novelty in a number of cases. For instance, the authors compare their ensemble predictions to 'single structure' predictions, which is a reasonable start, but it would be far better to test their method against *other ensemble-based methods*, a number of which have been around for quite some time (Proteins. 2007 Feb 1;66(2):399-421 ; J. Chem. Inf. Model., 2014, 54 (7), pp 2127-2138... etc.).

Response: Your point is well taken. We include more discussion of existing ensemble-docking methods:

“To test whether considering proteins' hidden states is crucial for predicting their functions, we developed a technique called Boltzmann docking that approximates compounds' relative binding affinities by calculating the ensemble-average score across a set of structural states, weighting each state by its equilibrium probability. Our approach differs from existing ensemble-docking methods, which dock compounds against a set of structures sampled via molecular dynamics simulations and then rank the compounds based on the highest score against any of the target structures.³³⁻³⁵ Boltzmann docking is more similar to methods that dock compounds against multiple conformations from NMR or crystal structures and weight the score against each structure by their relative contribution to the experimental signal.³⁶”

We also analyzed our docking data with a more classical ensemble approach by using only the highest scoring poses without weighting based on their populations. This led to a lower correlation with experimental data ($R = -0.02 \pm 0.07$) when compared with our Boltzmann docking method ($R = 0.30 \pm 0.10$). This analysis is now included in the main text:

“Boltzmann docking of cefotaxime against each of our variants represents a vast improvement over docking against single structures (Fig. 2b, $R = 0.30 \pm 0.10$ versus -0.37 ± 0.07). For comparison, we also performed classical ensemble docking, which ranks compounds based on their highest score against any single conformation from a set of structures. We find a correlation coefficient between ensemble docking and experiments of $R = -0.02 \pm 0.07$, indicating no correlation. Thus, while ensemble docking is superior to docking against single structures, which shows an anticorrelation, population-based averaging with Boltzmann docking is the only method that results in a physically meaningful correlation.”

On the Biological side, the novelty is also not quite what is suggested by the authors. For one thing, the 'hidden' conformations are indeed 'hidden' in conventional crystallography, but can be detected and characterized by biophysical NMR (particularly CPMG relaxation dispersion: Nature 438, 117-121), H/D exchange (particularly millisecond H/D exchange for loops: Lab Chip, 2013,13, 2528-2532) and even FPOP in some cases. This isn't to say that it's unhelpful to have computational methods that predict these conformations from crystal structures, but the fact remains that these 'hidden' conformations are not in fact invisible to experimentalists. Directly measuring changes in omega loop dynamics in various omega loop mutants, for instance, would be fairly straightforward experimentally, and would lead presumably to the same biological conclusion that is presented in this work. (Restricted omega loop dynamics = higher activity)

Response: We agree there are many valuable experimental techniques for detecting hidden conformations. Our intent was to demonstrate how MSMs can guide the application of these techniques. We have included additional citations and clarified in the text that there are other ways to detect hidden conformations:

“The combination of thermodynamic, kinetic, and structural information from MSMs can provide mechanistic insight and guide the application of experimental techniques^{23,24,31} for identifying hidden conformations. We reasoned MSMs could reveal hidden conformations that determine TEM specificity but that are not apparent from single “snap-shot” structures. First, we integrate our MSM with computational docking to develop a new approach called Boltzmann docking. Then, we test the MSM’s predictions of conformational heterogeneity using Fast Photochemical Oxidation of Proteins (FPOP), a chemical-footprinting technique. Finally, to determine the importance of hidden states, we design new variants to control their populations and then measure their activities *in vitro* and *in vivo*.”

The authors also claim that 'crystal structures of TEM variants do not provide a clear explanation of their functional parameters'. In fact, for the mutants being studied, there seems to be a consensus, based on crystallographic data that cephalosporinase activity-enhancing mutations act by 'opening' the active site (this is partly acknowledged by the authors), which strikes me as being how a crystallographer would describe the 'pinning' effect that is reported in the current work. Having said this, the computational model does provide additional indications as to the mechanism by which the active site is 'opened'.

Response: Our model contrasts with the consensus mechanism you describe. Rather than opening the active site, we find that G238S and E104K/R keep the active site closed. We have elaborated in the text how our model departs from existing models for these mutations (below). Additionally, it is entirely possible that other cephalosporinase activity-enhancing mutations have different mechanisms.

“Both of these substitutions have been studied extensively, but to our knowledge we are the first to present a model linking either mutation to restricted motion in the Ω -loop. Crystal structures of G238S-containing variants present conflicting models for how the 238-loop and Ω -loop interact. The TEM-52 structure lacks hydrogen bonds between the two loops⁷, but other structures capturing multiple conformations of the Ser238 side chain show it can hydrogen bond to the backbone carbonyls of Asn170 or Glu171.¹¹ We also observe formation of a hydrogen bond between Ser238 side chain and N170 backbone, but the precise conformation of the 238-loop differs. The crystal structures show that the 238-loop moves 2-4 Å away from the Ω -loop (measured by the C $_{\alpha}$ positions of Glu240 in G238S-containing structures versus TEM-1), which results in a small widening of the active site. In contrast, we observe that both loops assume conformations more similar to the wild-type crystal structure in our most populated cefotaximase states (Fig. 4). Our model suggests that rather than opening up the active site, G238S and E104K maintain a closed active-site architecture by pinning down the Ω -loop. In the absence of these substitutions, the Ω -loop exhibits greater flexibility and accesses conformations that increase its solvent exposure. This opening event in wild-type is a more dramatic change than the small widening caused by movement of the 238-loop in G238S-containing crystal structures. The models are not necessarily mutually exclusive, but ours challenges the proposal that wild-type cannot degrade cefotaxime because it is simply too big to fit in the active site. Our Boltzmann docking analysis supports our model because the structures with the highest docking score against cefotaxime are similar between wild-type and E104K/G238S (Fig. 3). Both docked structures have loop conformations resembling the wild-type crystal structure (Fig. 3, insets). We do not observe active-site opening when cefotaxime is bound, and the larger substrate can, indeed, fit in the wild-type active site without steric conflict. It is the fact that this binding-competent state is not highly populated in wild-type that underlies its low activity against cefotaxime. Finally, kinetic analysis of cefotaximase variants shows that higher catalytic efficiencies likely result from lower K_m values. Because acylation is the rate-limiting step, K_m approximates the substrate binding affinity.⁴⁴ Cefotaximase variants also have lower K_m s for benzylpenicillin (Table 1), which suggests restricting motion in the Ω -loop helps the enzyme bind both substrates. The low k_{cat} values observed for these variants might indicate that Ω -loop flexibility is

important for another step in the catalytic cycle, such as product release. We propose that the conformation captured by the wild-type crystal structure, particularly of the Ω -loop, is the binding-competent state for a diverse set of substrates, and that E104K/G238S stabilizes this state to allow for more promiscuous binding.”

C. The data are generally of high quality. There is some question of the appropriateness of the FPOP method for measuring loop dynamics, since it is seen largely as being sensitive to solvent accessibility, but it is possible that 'pinning' effect would restrict solvent access to the loop somewhat.

Response: We used the phrase loop “dynamics” to describe conformational heterogeneity rather than rates of exchange. Indeed we see that in wild-type, the omega-loop accesses conformations with greater solvent accessibility. FPOP allows us to detect these conformations and the results are consistent with the MSM predictions. We clarified in the text:

“Our model suggest that rather than opening up the active site, G238S and E104K maintain a closed active-site architecture by pinning down the Ω -loop. In the absence of these substitutions, the Ω -loop exhibits greater flexibility and accesses conformations that increase its solvent exposure.”

“Because the Ω -loop mobility we observe in MSMs results in significant changes in solvent accessibility (Fig. 4), we can experimentally test our insights using the FPOP approach. This technique is a chemical-footprinting method that reports on structural fluctuations by labeling solvent-exposed side chains with hydroxyl radicals and detecting the oxidized peptides with mass spectrometry.⁴⁵”

D. The authors did test their hypothesis against a significant and sufficient number of mutants in my view. However it would have been nice to see more *sites* since this entire region is known to undergo mutations that affect specificity: J Bacteriol. 1996 Apr;178(7):1821-8. Moreover, many of those mutations are non-specific, suggesting that simply increasing the dynamics of the omega loop (as opposed to pinning it) also causes an increase in cephalosporinase activity.

Response: Thank you for the suggestion. We are familiar with this paper and other studies that focus on mutations in the omega-loop but have not yet investigated them ourselves. We believe our methods could be applied to these mutations in the future and possibly even predict mutations elsewhere that have not been identified.

E. The main conclusion - that ensemble-based docking can provide enhanced functional predicitions and insights for enzymes is mostly supported by the data (although, it should be noted that there are many enzyme systems for which the 'hidden' intermediates are more directly linked to the catalytic reaction coordinate than substrate binding. The TEM-1 / Omega-loop system happens to be ideal for 'catalytic efficiency' effects that are mostly derived from substrate binding enhancement)

Response: We agree that docking might not be the best approach for all enzymes. We clarify in the text that we use substrate docking as an imperfect proxy for activity (below). We also added an explanation for why we think using cefotaximase states from the MSM outperforms Boltzmann docking by suggesting that it captures catalytically relevant states not participating in substrate binding. Although it requires more information (ie known variants with alternative substrate specificities), we expect this method to be more generalizable than Boltzmann docking.

“Boltzmann docking performed better than docking against single structures by accounting for conformational heterogeneity, but any docking approach is limited by using substrate binding as a proxy for activity. Enzymes undergo conformational changes throughout a catalytic cycle in concert with chemical transformations to the substrate, so these states would not necessarily score well. MSMs, on the other hand, may capture these catalytically relevant states. Therefore, a powerful approach to classify the activities of new variants is to compare MSMs of their structural ensembles to MSMs for variants with known functions.”

F. Only suggestion would be a more thorough investigation of different mutation sites in the omega loop, but I do believe the number of mutants currently looked at is sufficient.

Response: Thank you for the suggestion, and we intend to study these sites in the future.

G. The references are largely adequate, but should include some discussion and references to previous work in ensemble-based docking. Wouldn't hurt to cite experimental methods for detecting and characterizing 'hidden' conformations as well.

Response: We have included additional citations and clarified in the text that there are other ways to detect "hidden" conformations:

"The combination of thermodynamic, kinetic, and structural information from MSMs can provide mechanistic insight and guide the application of experimental techniques^{23,24,31} for identifying hidden conformations."

H. The clarity of writing was good.

Response: Thank you!

Reviewer #3 (Remarks to the Author):

In this manuscript, the authors nicely combined MSMs with biochemical and mass spectrometry footprinting experiments to successfully elucidate how different variants of TEM β -lactamase (TEM) can lead to their dramatically functional differences. This cannot be explained by static crystal structures. The MSMs constructed from MD simulations not only facilitated the Boltzmann docking which outperforms the single-structure docking, but also identified the unprecedented hidden states of TEM responsible for hydrolyzing cefotaxime. The flexibility of Ω -loop predicted by MSMs was confirmed by mass spectrometry footprinting. Furthermore, the author used MSMs together with measurements of enzyme activity and MIC to show that both electrostatic interactions and hydrophobic surface area could have led to stronger enzyme activity.

This work holds great promise to facilitate the development of new drugs to fight against antibiotic resistance. Using TEM as an example, the authors shed new light on the structural and molecular basis of cefotaxime resistance. The methodology (MSM-guided mass spectrometry footprinting and functional assays) can serve as a powerful means for investigating structural basis of other enzyme functions. That being said, I also have a few questions and suggestions concerning the details of this manuscript. Therefore, I would like to recommend its publication at Nature Communications after the following comments are addressed:

1. The authors showed that a few hidden states (e.g. in Fig. 2C) are important for predicting cefotaxime activity. I am curious how did the authors select these important hidden states from their MSM containing 1000 states? I would also suggest the authors to display and discuss representative structures from these hidden states.

Response: We selected the states by identifying which were preferred by the known cefotaximase variant but not by the wild-type sequence (cefotaximase states) and visa versa (non-cefotaximase states). The specific procedure for doing this was added the Methods (below).

“MSMs for comparing the structural preferences of different variants were constructed based on the same set of active site residues. First, every 100th data point from simulations of each variant were pooled together and clustered into 1,000 states with a k-medoids algorithm. Then the equilibrium probability of each state for a given variant was calculated using the same approach described before using just the data for that variant. Using a common set of states to describe the thermodynamics and kinetics of each variant provides a basis for directly comparing the probabilities that different variants will adopt a given conformation. Of the 1,000 states in this combined state space, we found that 350 of these states have higher populations in E104K/G238S than wild-type. As explained in the text, we designate these states as cefotaximase states and employ the sum of the equilibrium populations of these states (with error bars from bootstrapping, as explain below) as a surrogate for cefotaxime activity. Another 399 states are more populated by wild-type than E104K/G238S, which we designate as non-cefotaximase states.”

We have added a new main text figure (Fig. 4, below) to show the most highly populated of these states and included more discussion of the structures (below).

“Both of these substitutions have been studied extensively, but to our knowledge we are the first to present a model linking either mutation to restricted motion in the Ω -loop. Crystal structures of G238S-containing variants present conflicting models for how the 238-loop and Ω -loop interact. The TEM-52 structure lacks hydrogen bonds between the two loops⁷, but other structures capturing multiple conformations of the Ser238 side chain show it can hydrogen bond to the backbone carbonyls of Asn170 or Glu171.¹¹ We also observe formation of a hydrogen bond between Ser238 side chain and N170 backbone, but the precise conformation of the 238-loop differs. The crystal structures show that the 238-loop moves 2-4 Å away from the Ω -loop (measured by the C $_{\alpha}$ positions of Glu240 in G238S-containing structures versus TEM-1), which results in a small widening of the active site. In contrast, we observe that both loops assume conformations more similar to the wild-type crystal structure in our most populated cefotaximase states (Fig. 4). Our model suggests that rather than opening up the active

site, G238S and E104K maintain a closed active-site architecture by pinning down the Ω -loop. In the absence of these substitutions, the Ω -loop exhibits greater flexibility and accesses conformations that increase its solvent exposure. This opening event in wild-type is a more dramatic change than the small widening caused by movement of the 238-loop in G238S-containing crystal structures. The models are not necessarily mutually exclusive, but ours challenges the proposal that wild-type cannot degrade cefotaxime because it is simply too big to fit in the active site.”

Figure 4. A crystal structure of TEM-1 (blue, PDB 1BTL) is overlaid with the two most populated structures taken from MSMs of the (a) non-cefotaximase states (green), which are favored by wild-type and the (b) cefotaximase states (orange), which are favored by the E104K/G238S. (c) Large structural rearrangements in the Ω -loop distinguish low-energy non-cefotaximase states from cefotaximase states, which more closely resemble the conformation captured by the crystallographic structure. Residues 104 and 238 are shown in spheres.

2. This is a related question on the hidden states. In Fig. 2C, the authors show strong correlations between the populations of hidden states with the experimental catalytic rates. If the Boltzmann weighted docking scores to these hidden state conformations rather than their populations are displayed, I am wondering if the same or even better correlation will hold? Anyway, I would suggest that authors to include some discussions on docking to these hidden states.

Response: Restricting our analysis to the cefotaximase states would give an equivalent result. Since the MSMs have a shared state space that is defined in terms of shared active site residues (all the residues that can interact with substrate), the docking scores against these states will be the same for all variants. All the differences between the variants arise because of the different populations of these states.

3. In the Boltzmann docking, docking scores from all MSM states are weighted averaged. While in typical ensemble docking studies, only top docking poses are selected, while others are discarded. If the initial binding to one particular can be significantly stabilized by subsequent induced fit, I would think the ensemble docking approach might work better. Could the authors add in some comment on the pros and cons of these two different approaches?

Response: We also analyzed our docking data with a more classical ensemble approach by using only the highest scoring poses without weighting based on their populations. This led to a lower correlation with experimental data ($R = -0.02 \pm 0.07$) when compared with our Boltzmann docking method ($R = 0.30 \pm 0.10$). This analysis is now included in the main text:

“To test whether considering proteins’ hidden states is crucial for predicting their functions, we developed a technique called Boltzmann docking that approximates compounds’ relative binding affinities by calculating the ensemble-average score across a set of structural states, weighting each state by its equilibrium probability. Our approach differs from existing ensemble-docking methods, which dock compounds against a set of structures sampled via molecular dynamics simulations and then rank the compounds based on the highest score against any of the target structures.³³⁻³⁵ Boltzmann docking is more similar to methods that dock compounds against multiple conformations from NMR or crystal structures and weight the score against each structure by their relative contribution to the experimental signal.³⁶”

4. The Markovian lag time of their MSM is at over 1ns, while the authors use 10ps as the lag time to construct their MSM to calculate equilibrium populations. I agree with the authors that thermodynamics generally reaches convergences faster than kinetics. To make it more convincing, I would suggest the authors to calculate the equilibrium population with different lag times (e.g. 20ps) and show the invariance of obtained equilibrium populations.

Response: We agree this is an important point. We have verified that we get the same results for lag times ranging from 10 ps to 10 ns. For example, the correlation coefficient between populations from an MSM with a lag time of 10 ps and a model with a lag time of 10 ns is greater than 0.999. The most populated state, regardless of lag time, is state 361 and its equilibrium probability ranges from 0.0286 with a lag time of 10 ps to 0.0288 for a lag time of 10 ns. We have added the following to the Methods:

“Consistent with past work demonstrating that thermodynamics converge far more quickly than kinetics,⁶⁴ the thermodynamics of our models are insensitive to varying the lag time from 10 ps to 10 ns, so equilibrium populations of each state were determined by calculating a matrix of transition probabilities between every pair of states with the transpose method and a lag time of 10 ps and solving for the normalized left-eigenvector of this matrix.”

5. The following paper discussed an approach of using MSM followed by protein-RNA docking to elucidate the miRNA-Ago recognition mechanisms, which could be relevant to the current work when the authors work on further elucidating the molecular recognition mechanisms:

Jiang, H., Sheong, F.K., Zhu, L., Gao, X., Bernauer, J., Huang, X., PLoS. Comp. Bio., 11(7): e1004404, (2015)
Gu, S., Silva, D.A., Meng, L., Yue, A., Huang, X., PLoS. Comp. Bio., 10(8):e1003767, (2014)

Response: Thank you. We have added one of your suggested citations but could not fit the other one due to reference restrictions.

6. The plots in Figure 2.b shows that Boltzmann docking outperforms the single-structure docking in predicting the specificity of TEM variants. It would be helpful if the docked structures accompany these plots to show the structural details of TEM-cefotaxime interactions.

Response: Thank you for the suggestion. We have added a new figure to highlight these structural details (Fig. 3).

Figure 3. Highest ranking poses from Boltzmann docking of cefotaxime against (a) wild-type (green) and (b) E104K/G238S (orange). Both structures are shown overlaid with the crystal structure of TEM-1 (blue, PDB ID 1BTL) for reference. Cefotaxime is in cyan. Insets depict key loops flanking the active site with the catalytic Ser70 shown in red.

7. It seems that the correlation coefficients (R) for Supplementary Figure 1. a&b are missing.

Response: Thank you for your close reading. We have fixed this error.

Reviewer #4 (Remarks to the Author):

The manuscript "Modeling proteins' hidden conformations to predict antibiotic resistance" describes a combined computational-experimental study of the role of conformational heterogeneity stabilized by mutations in determining TEM beta-lactamase (TEM) specificity for beta-lactam antibiotics. This is surprising, considering the traditional viewpoint that TEM is a rather rigid protein. Despite this, the ability of TEM to access alternative conformations can explain how mutations far from the substrate binding site can alter substrate specificity by up to several orders of magnitude.

While there are some weak statistical components of the manuscript---namely, a lack of statistical rigor in correlation coefficients, and claims made about Boltzmann docking that are almost certainly artifacts of the poor statistical power of $N = 12$ data points---the overall story is extremely interesting, and would be of great interest to the readership of Nature Communications. Taken as a whole, the characterization of hidden states by molecular simulation, the footprinting experiments, and the engineered mutants do seem to suggest a significant role for these "hidden states" in catalysis.

I recommend publication after fixing the statistical deficiencies and other issues noted below.

Issues that should be addressed before the manuscript is acceptable for publication:

* All correlation coefficients R should incorporate some form of uncertainty estimate arising from the use of a finite dataset.

Otherwise, there is no way to evaluate whether statements such as " $R = 0.35$ versus $R = 0.07$ " carry any statistical significance.

At minimum, this should be bootstrapped statistical errors, or better yet, confidence intervals.

The authors are referred to <http://dx.doi.org/10.1007%2Fs10822-014-9753-z> for a more detailed description of the appropriate statistics for R in molecular modeling.

Response: The fact that we can design successful experiments based on our models, even in the face of statistical uncertainty, shows that there is something to the trends we present and greatly strengthens our conclusions. But we agree that statistical rigor is important and have added errorbars, as explained below:

"Error bars on the population of a subset of MSM states were obtained via bootstrapping. That is, we drew 100 independent subsamples of the data (with replacement) and reported the mean and standard deviation of the sum of the equilibrium probabilities of the subset of states of interest. Correlation coefficients between the populations of subsets of MSM states and experiments were obtained by calculating the Pearson correlation coefficient between all n possible subsets of $n-1$ data points (where n is the number of TEM variants) and reporting the mean and standard deviation of these correlation coefficients."

* The quantities being correlated (line 94) are not clearly stated.

Response: We have added the following clarification:

"Docking against single structures of each variant completely fails to predict their abilities to degrade cefotaxime, as measured by their k_{cat}/K_m values."

* The discussion attributing failures of docking solely to inadequacies in the forcefield (lines 93-103) is unfairly shortsighted. Both the fact that docking scores a single structure (rather than computing the free energy of binding by integrating over conformational space) and the neglect of protonation/tautomeric states could also be contributors, as could deficiencies in sampling within the conformations (such as sidechain conformational sampling). Even the manner in which the protein structure was prepared for docking could contribute. This experiment is really only useful in comparison with the next experiment in which everything else is kept constant except weighted contributions from multiple conformational states are included.

Response: We agree completely, as stated in the text:

"These failed predictions could be due to shortcomings in the force field, which

describes the atomic interactions used to produce a docking score. However, they could also be interpreted as evidence for the inadequacy of focusing on single, rigid structures when we know that proteins actually adopt a distribution of different structures at thermal equilibrium. Indeed, we have previously demonstrated that TEM β -lactamase adopts a range of different conformations.^{12,32} Therefore, we wanted to explore whether inadequacies in the force field are really to blame for the failures of docking, or if inadequate accounting for proteins' hidden states is responsible."

* The manner in which the "representative structure" from each MSM state was selected---and the manner in which the MSM states were selected---should be stated. Are these microstates or lumped macrostates? How many were there to choose from?

Response: We have added the following to the methods:

"This procedure resulted in the following number of clusters for each variant: 2046 for wild-type, 1891 for G238S, 1693 for E104K, 1280 for E104K/G238S, 1467 for E104R, 942 for E104R/G238S, 1338 for E104A, 1206 for E104A/G238S, 1530 for E104D, 2525 for E104D/G238S, 1090 for E104M, 780 for E104M/G238S, 1358 for E240K/E104K, 2113 for R164D/G238S, and 2812 for R164E/G238S. We selected the cluster centers for each state as representative structures. We used the representative structures for each cluster as the basis for our Boltzmann docking approach. Another alternative would be to coarse-grain the model by merging clusters into macrostates. However, doing so could easily merge geometrically distinct conformations and lead to inaccurate estimates of the total probability of binding competent conformations."

* I count 12 measurements being depicted in Figure 2b. Using Eq. 54 of [<http://dx.doi.org/10.1007%2Fs10822-014-9753-z>] to compute the minimum R value outside of the 95% confidence interval for no correlation gives a minimum R threshold of 0.57. That means that, with only $N = 12$ data points, the 95% confidence interval for R for uncorrelated data ($R = 0$) includes all R values up to 0.57. Therefore, there is insufficient data to state that the improved correlation achieved by Boltzmann docking ($R = 0.35$ vs $R = 0.07$) is due to anything but chance.

Response: As noted above, we appreciate the importance of statistical rigor but want to emphasize the importance of our experimental results for supporting the utility of our approaches. We have added the following to the main text to acknowledge this point:

"Despite the apparent trend in correlation coefficients as one moves from docking against single structures, to ensemble-docking, and then to Boltzmann docking, we acknowledge that we cannot discount the possibility this trend is due to chance given the number of data points we have.³⁸ Furthermore, while Boltzmann docking appears to outperform alternative methods, the absolute correlation between Boltzmann docking and experiments is only moderate. It does suggest, however, that our MSMs contain information that is not encoded in single structures. We reasoned that our MSMs could contain information about the catalytic cycle beyond just the substrate-binding affinity, and next sought to learn what insights further analysis of these models might provide."

* "We queried the MSMs for states that are significantly more populated by one sequence over the other." Can you clarify whether microstates (which are simply configurational clusters) or lumped macrostates (which might represent kinetically meaningful conformations) are being referred to here?

Response: We have added the following to the methods:

"We selected the cluster centers for each state as representative structures. Another alternative would be to coarse-grain the model by merging clusters into macrostates. However, doing so could easily merge geometrically distinct conformations and lead to inaccurate estimates of the total probability of binding competent conformations."

* Why should the docking score correlate with k_{cat}/K_m and not $\log(k_{cat}/K_m)$ or $\log(K_m)$? The docking score should be on an energy ($\log K_d$) scale, while the measured k_{cat} , K_m , or k_{cat}/K_m are on exponentiated energy scale.

Response: Your point is well taken. We have re-analyzed the data using $\ln(k_{cat}/K_m)$ to compare with the docking scores, and found our conclusions are unchanged. Correlation with our Boltzmann docking score is within error of our first value ($R = 0.30 \pm 0.10$), while there is a puzzling anticorrelation for docking against static structures ($R = -0.37 \pm 0.07$), which only emphasizes our point. We have updated the text and Fig. 2 (below) to reflect these changes.

Figure 2. An ensemble perspective predicts the effects of mutations on TEM β -lactamase's specificity better than single structures. (a) Docking scores for cefotaxime against single-structure homology models of each variant have no meaningful correlation ($R = -0.37 \pm 0.07$) with the measured catalytic efficiencies ($\ln(k_{cat}/K_m)$). (b) There is a correlation ($R = 0.30 \pm 0.10$) between the Boltzmann-docking scores for cefotaxime and the measured catalytic efficiencies ($\ln(k_{cat}/K_m)$). (c) There is a stronger correlation ($R = 0.74 \pm 0.11$) between the predicted populations of cefotaximase states and the measured catalytic efficiencies (k_{cat}/K_m). Double mutants are shown in blue, and single mutants are shown in red. Error bars are standard errors from the fit. Computational errors, which are less than 3×10^{-3} , are not shown for clarity.

* "populations of our hidden cefotaximase states" - how was this quantity defined? I couldn't seem to find a quantitative definition of how this was computed---was this just microstates with higher populations in WT as opposed to mutant? How robust is that set to statistical error in the microstates? Interestingly, this correlation coefficient ($R = 0.83$) is large enough to likely be significant!

Response: We have added the following to the methods:

“Of the 1,000 states in this combined state space, we found that 350 of these states have higher populations in E104K/G238S than wild-type. As explained in the text, we designate these states as cefotaximase states and employ the sum of the equilibrium populations of these states (with error bars from bootstrapping, as explain below) as a

surrogate for cefotaxime activity. Another 399 states are more populated by wild-type than E104K/G238S, which we designate as non-cefotaximase states.

Error bars on the population of a subset of MSM states were obtained via bootstrapping. That is, we drew 100 independent subsamples of the data (with replacement) and reported the mean and standard deviation of the sum of the equilibrium probabilities of the subset of states of interest.”

* Molecular dynamics simulations: This section is lacking sufficient detail for a competent practitioner to reproduce the simulations.

Response: We have added the following to the methods:

* "with a position restraint on all 300 protein heavy atoms" - what spring constant was used?

1,000 kJ/mol/nm²

* what temperature and pressure was used?

300K and 1 bar

* was a long-range isotropic dispersion correction used?

A long-range dispersion correction was employed for both energy and pressure.

* which sites were turned into virtual sites?

for all hydrogens

* what PME parameters were used (grid spacing, order, error tolerance)?

with a grid spacing of 0.12 nm, PME order 4, and tolerance of 1e-5

* what v-scale interval?

0.1 ps

* How often were snapshots written?

Snapshots were stored every 10 ps.

* MSM construction and analysis:

Response: We have added the following to the methods:

* Which version of MSMBuild was used?

V2.8

* How many resulting microstates were there?

“This procedure resulted in the following number of clusters for each variant: 2046 for wild-type, 1891 for G238S, 1693 for E104K, 1280 for E104K/G238S, 1467 for E104R, 942 for E104R/G238S, 1338 for E104A, 1206 for E104A/G238S, 1530 for E104D, 2525 for E104D/G238S, 1090 for E104M, 780 for E104M/G238S, 1358 for E240K/E104K, 2113 for R164D/G238S, and 2812 for R164E/G238S.”

* Was any lumping performed, or were the microstates used as "states" throughout?

“We selected the cluster centers for each state as representative structures. We used the representative structures for each cluster as the basis for our Boltzmann docking approach. Another alternative would be to coarse-grain the model by merging clusters into macrostates. However, doing so could easily merge geometrically distinct conformations and lead to inaccurate estimates of the total probability of binding competent conformations.”

* Supplementary Figure 4: This plot is lacking error bars, and the x- and y-axes are both lacking units.

Response: We added error bars and moved the units from the title to the axis labels for clarity. The methods section explains:

“Implied timescales were also obtained by calculating the implied timescales of all n possible subsets of n-1 trajectories (where n is the number of independent simulations) and reporting the mean and standard deviation of these timescales.”

* Supplementary Table 1: I'm not sure this table satisfies the NPG Statistical Guidelines [<http://www.nature.com/srep/publish/guidelines>] in its current form.

Response: We note that for the data in the table:

“MIC determination was repeated at least three times. Values are most commonly observed.”

While this is not a measure of statistical uncertainty, it is the standard practice for quoting “error” in MIC measurements, as outlined by the Clinical and Laboratory Standards Institute guidelines (CLSI, in Methods for Dilution Antimicrobial Susceptibility Tests for Bacteria That Grow Aerobically; Approved Standard- Ninth Edition, CLSI document M07-A9, Clinical and Laboratory Standards Institute, Wayne, PA, 2012.)

Reviewers' Comments:

Reviewer #1 (Remarks to the Author)

Major points:

The change in the underlying data for Fig 2 shows a strong negative correlation for the ground states. Notably, this correlation is more significant than the Boltzmann docking one. While the Boltzmann docking is still an improvement over the previous static docking and gets the "sign" of the correlation right - this also suggests a different model. Perhaps the ground states ARE anticorrelated with binding?! This possibility should at least be discussed with reference to why the MSMs favor or disfavor the explanation.

Why isn't 2C on log scale for correlation/graphing? The strong correlation there seems driven by the one outlier point? They should clarify.

Minor points:

* In general - the backbone structures shown in the figures show that the active site can be structured into a cefotaximase-accomodating state. But the backbones alone do not provide a good explanation for the energetic differences that distinguish "cefotaximase" states. Show the side chains? What are the key interactions that are important here? And label the figures. Please.

* Please add residue labels to Fig. 1 -- it's very hard to tell which residues are which. Multiple views should be provided to give an idea of the similarity of the active site loops. The ground state measurements for relevant distances should also be plotted on Fig S1 for clarity.

Reviewer #2 (Remarks to the Author)

The authors appear to have adequately addressed the comments from the previous review. The manuscript is now publishable in my view, with the caveat that for me this work (while unquestionably well done and interesting) may not quite meet the very high 'impact' bar for Nature Comms. However, that is ultimately a call for the editors. From a scientific point of view, the paper is ready.

Reviewer #3 (Remarks to the Author)

The authors have appropriately addressed my comments in the revised manuscript and I would recommend its publication at Nature Communications.

Reviewer #4 (Remarks to the Author)

I have reviewed the 21 page (!!!) detailed response to referee critiques the authors have provided with their revised manuscript. In their response, the authors do an appropriate job of addressing the statistical concerns I raised in my earlier critique. While the statistical significance of some of the conclusions is still weak as a result of the small number of mutations tested, the results are presented

in their appropriate statistical context, and the resulting discussion has been appropriately tempered.

The overall body of work remains provocative and of broad interest. I happily give my support for rapid publication of this work.

Reviewer #1 (Remarks to the Author):

Major points:

The change in the underlying data for Fig 2 shows a strong negative correlation for the ground states. Notably, this correlation is more significant than the Boltzmann docking one. While the Boltzmann docking is still an improvement over the previous static docking and gets the "sign" of the correlation right - this also suggests a different model. Perhaps the ground states ARE anticorrelated with binding?! This possibility should at least be discussed with reference to why the MSMs favor or disfavor the explanation.

Response: It is true that an anticorrelation is, in fact, a correlation. It's possible our docking values are serving as a proxy for product affinity (rather than substrate affinity), which might explain the anticorrelation. However, substrate binding is certainly a requirement for turnover, but when you extrapolate the anticorrelation to very low binding affinities, the model predicts efficient turnover. Since our goal is to predict activity against novel compounds, an anticorrelation is of limited utility. Furthermore, given the positive correlation observed for our Boltzmann docking method, we think the anticorrelation likely reflects shortcomings of docking against single structures. We have added our reasoning to the text:

"Docking against single structures of each variant fails to predict their abilities to degrade cefotaxime, as measured by their k_{cat}/K_m values. In fact, we observe that the docking scores and activities are anticorrelated ($R = -0.37 \pm 0.07$, Fig. 2a). While it is possible this anticorrelation suggests an alternative model for TEM catalysis, it is more likely that it simply highlights the limitations of docking against single structures. For example, this model would incorrectly predict that compounds with no binding affinity would be excellent substrates."

As discussed later in the text, our MSM method for assessing the populations of cefotaximase states is agnostic to their role in the mechanism. Therefore, it is not limited by our earlier hypothesis that substrate binding is a reasonable surrogate for activity.

Why isn't 2C on log scale for correlation/graphing? The strong correlation there seems driven by the one outlier point? They should clarify.

Response: We have changed Fig. 2C so that both axes are on a log scale for clarity (below). Our original rationale was that docking scores are on an energy scale and should be compared to the logs of rates, while MSM populations are the exponentials of energies and should be compared directly to rates. However, we agree that putting all three plots on an energy scale makes the reader's job easier. After switching our results to a log scale, we find the correlation coefficient between our computational models and experiments is 0.79 ± 0.03 . Leaving out the outlier, E104K/G238S, results in a correlation coefficient of 0.74 ± 0.04 , showing that our correlation is not solely dependent on this single data point. In making this update, we realized that we had plotted an older version of our results (based on a smaller subset of atoms defining the active site). Our updated analysis leaves the key results unchanged. We've amended the text (changes are highlighted) as follows:

"We first constructed MSMs for our designed variants and then assessed the degree to which they populate the cefotaximase states. We then experimentally measured their *in vitro* activities against cefotaxime and the extent to which they confer cefotaxime resistance to *E. coli* (Table 1). Given that similar trends exist in the single and double mutants, we focus on the variants containing the sensitizing G238S mutation. As predicted, E104D/G238S has a similar probability of adopting cefotaximase states as G238S alone and also has similar activity against cefotaxime. E104R/G238S, E104M/G238S and E104A/G238S all populate the cefotaximase states more than G238S alone, as predicted. However, in contrast with expectations based on charge arguments alone, E104R/G238S does not populate these states as extensively as

E104K/G238S (Fig. 2c). These results are consistent with both *in vitro* and *in vivo* experiments, which show E104R/G238S degrades cefotaxime better than G238S but not as well as E104K/G238S (Table 1). Comparing the conformations adopted by E104R/G238S to those of E104K/G238S reveals that Arg104 interacts more strongly with residues 170 and 171 in the Ω -loop, displacing key interactions with the catalytic water⁴² in the active site and reducing activity against both benzylpenicillin and cefotaxime (Supplementary Fig. 3). Consistent with our hypothesis that van der Waals interactions play an important role in pinning position 104 to the Ω -loop, E104M confers greater cefotaximase activity than does E104A. In fact, in the wild-type background E104M has greater activity than either of the positively-charged variants. Positive epistasis between G238S and E104K, however, results in this double mutant surpassing all others in cefotaximase efficiency. Taken together, these variants imply that both charge and hydrophobic surface area contribute to rate enhancement.

Our approach also tends to successfully identify variants that do not have significant cefotaxime activity. Based on our intuition about the importance of electrostatic interactions, we designed three additional variants intended to favor the closed Ω -loop conformations important for cefotaximase activity. For example, we reasoned that introducing negatively-charged residues behind the acidic Ω -loop at position 164 might increase cefotaxime activity by favoring closed conformations through electrostatic repulsion. However, MSMs for R164E/G238S predicts this variant does not populate the cefotaximase states as much as other known or predicted cefotaximases, and these predictions are consistent with both *in vitro* and *in vivo* experiments (Table 1). R164D/G238S is predicted to populate cefotaximase states more than R164E/G238S but less than known cefotaximases, which is consistent with its measured activity. We also tried changing position 240 to a positively-charged residue with the E104K/E240K variant in an attempt to favor closed Ω -loop conformations through electrostatic attraction. In this case, our MSM suggested this variant could have activity against cefotaxime, but this prediction was not borne out experimentally (Table 1). Despite this exception, the general agreement between our models and experiments demonstrates the added power of MSMs over biochemical intuition alone.

Impressively, the populations of our hidden cefotaximase states provide the most accurate predictions of cefotaxime activity (Fig. 2c, $R = 0.79 \pm 0.03$, compared to $R = 0.30 \pm 0.10$ for Boltzmann docking and $R = -0.37 \pm 0.07$ for docking against single structures). Furthermore, the correlation is robust to exclusion of the variant used to define cefotaximase states, E104K/G238S ($R = 0.74 \pm 0.04$). This result supports our conclusion that MSMs provide a reasonably accurate depiction of TEM β -lactamases' structural ensembles. Boltzmann docking performed better than docking against single structures by accounting for conformational heterogeneity, but any docking approach is limited by using substrate binding as a proxy for activity. Enzymes undergo conformational changes throughout a catalytic cycle in concert with chemical transformations to the substrate, so these states would not necessarily score well. MSMs, on the other hand, may capture these catalytically relevant states. Therefore, a powerful approach to classify the activities of new variants is to compare MSMs of their structural ensembles to MSMs for variants with known functions."

Figure 2. An ensemble perspective predicts the effects of mutations on TEM β -lactamase's specificity better than single structures. (a) Docking scores for cefotaxime against single-structure homology models of each variant are anticorrelated ($R = -0.37 \pm 0.07$) with the measured catalytic efficiencies ($\ln(k_{cat}/K_m)$). (b) There is a correlation ($R = 0.30 \pm 0.10$) between the Boltzmann-docking scores for cefotaxime and the measured catalytic efficiencies ($\ln(k_{cat}/K_m)$). (c) There is a stronger correlation ($R = 0.79 \pm 0.03$) between the natural logarithm

of the populations of cefotaximase states and the measured catalytic efficiencies ($\ln(k_{cat}/K_m)$). The correlation is robust to exclusion of E104K/G238S ($R = 0.74 \pm 0.04$). Double mutants are shown in blue, and single mutants are shown in red. Error bars are standard errors from the fit. Computational errors are less than 3×10^{-3} . The natural logarithms of MSM populations and experimentally determined catalytic efficiencies are reported to put these quantities on an energy scale, while docking scores are naturally on an energy scale.

Minor points:

* In general - the backbone structures shown in the figures show that the active site can be structured into a cefotaximase-accomodating state. But the backbones alone do not provide a good explanation for the energetic differences that distinguish "cefotaximase" states. Show the side chains? What are the key interactions that are important here? And label the figures. Please.

Response: The cefotaximase states depicted in Fig. 3 are representative structures. While the backbone conformation is similar between the cefotaximase states, the sidechain conformations can vary quite a bit. We only depict the backbone, because we didn't want to misrepresent the ensemble. However, the structures will be made available upon request to readers.

* Please add residue labels to Fig. 1 -- it's very hard to tell which residues are which. Multiple views should be provided to give an idea of the similarity of the active site loops. The ground state measurements for relevant distances should also be plotted on Fig S1 for clarity.

Response: We have revised Fig. 1 to include key residue labels and added a new Fig. S1 to show multiple views of the active site loops (below). We also have revised the original Fig. S1A, now Fig. S2A, to highlight which distance is being plotted (below).

Figure 1. Overlay of TEM-1 (blue, PDB 1BTL) and TEM-52 (yellow, PDB 1HTZ) reveals subtle conformational differences (heavy-atom RMSD = 0.60 Å), particularly in the loops containing the mutations, but very similar active-site architectures (inset, heavy-atom RMSD = 0.33 Å). Residues at positions 104 and 238 flank the active site and are shown in pink spheres. See also Supplemental Fig. S1.

Supplementary Figure 1. Overlay of important loops in the active sites of TEM-1 (blue, PDB 1BTL) and TEM-52 (yellow, PDB 1HTZ) reveals subtle conformational differences, particularly in the 238-loops.

Supplementary Figure 2. E104K and G238S substitutions restrict motion in the Ω -loop. (a) The distribution of distances between position 104 and 167 of the Ω -loop (inset) shows that states with a short distance are more probable in variants containing the E104K substitution. (b) A representative structure from simulations containing the G238S substitution reveals a hydrogen bond between the side chain of S238 and backbone carbonyl of N170.

Reviewer #2 (Remarks to the Author):

The authors appear to have adequately addressed the comments from the previous review. The manuscript is now publishable in my view, with the caveat that for me this work (while unquestionably well done and interesting) may not quite meet the very high 'impact' bar for Nature Comms. However, that is ultimately a call for the editors. From a scientific point of view, the paper is ready.

Reviewer #3 (Remarks to the Author):

The authors have appropriately addressed my comments in the revised manuscript and I would recommend its publication at Nature Communications.

Reviewer #4 (Remarks to the Author):

I have reviewed the 21 page (!!!) detailed response to referee critiques the authors have provided with their revised manuscript. In their response, the authors do an appropriate job of addressing the statistical concerns I raised in my earlier critique. While the statistical significance of some of the conclusions is still weak as a result of the small number of mutations tested, the results are presented in their appropriate statistical context, and the resulting discussion has been appropriately tempered.

The overall body of work remains provocative and of broad interest. I happily give my support for rapid publication of this work.